

# Challenges assessing the effect of AMVs to improve the predictability of a medicane weather event using the EnKF. Storm-scale analysis and short-range forecast.

Diego S. Carrió [1,2]

[1]School of Geography, Earth and Atmospheric Sciences, The University of Melbourne, Parkville, Victoria, Australia
[2]ARC Centre of Excellence for Climate Extremes, Australia

**Correspondence:** Diego Saúl Carrió (diego.carriocarrio@unimelb.edu.au)

**Abstract.** Coastal population in the Western Mediterranean basin is frequently affected by high-impact weather events that produce loss of life and property for an invaluable amount. Among the wide spectrum of maritime severe weather events, tropical-like Mediterranean cyclones (a.k.a. medicanes) draw particular attention, specially due to their poor predictability. The accurate prediction of this kind of events still remains a key challenge to our community, mainly because of (i) the errors

in the initial conditions, (ii) the lack of accuracy modelling micro-scale physic processes and (iii) the chaotic behavior inherent to current numerical weather prediction models. In particular, the 7th October 2014 Qendresa medicane, that took place over the Sicilian channel, affecting the Islands of Lampedusa, Pantelleria and Malta was selected for this study because of its extremely low predictability behavior in terms of its track and intensity. To enhance the prediction of Qendresa, a high-resolution (4-km) ensemble-based data assimilation technique, known as Ensemble Kalman Filter (EnKF) is used. In this study both *in-situ*

conventional and satellite-derived observations are assimilated with the main objective of improving Qendresa's model initial conditions and thus, its subsequent forecast. The performance of the EnKF system and its impact on the Qendresa forecast is quantitatively assessed using different deterministic and probabilistic verification methods. A discussion in terms of the relevant physical mechanisms adjusted by the EnKF is also provided. Results reveal that the assimilation of both conventional and satellite-derived observations improves the short-range forecasts of the trajectory and intensity of Qendresa. In this context,

it is shown the relevance of assimilating satellite-derived observations to improve the pre-convective estimation of Qendresa's upper-level dynamics, which is key to obtain a realistic track and intensity forecast of this event.

## 1 Introduction

Among the entire set of high-resolution observation platforms available, the assimilation of both ground-based radar [e.g., Snyder and Zhang (2003)] and satellite (Jones et al., 2015) observations have been demonstrated to be very valuable in order

to enhance the predictability of tropical cyclones, heavy precipitation events or even tornadoes. Several studies have shown the benefits of assimilating radar data and found that both radial velocity and reflectivity observations provide information leading to improvements of convective-scale analyses and their corresponding forecasts (Snyder and Zhang, 2003; Dowell et al., 2004; Tong and Xue, 2005; Dawson II et al., 2012; Aksoy, A. et al., 2009; Yussouf et al., 2013). In general, radar data assimilation





studies are focused on performing accurate analysis of already mature convective systems with the main aim of improving the

very short-range forecasts (i.e., minutes-few hours) of severe weather events, such as storm-scale tornado circulations. However, the correct representation of the pre-convective environment, which is key to obtain short-range (i.e., hours-few days) forecasts, has received somewhat less attention. Improving the time and location forecast of convection initiation has demonstrated to be a significant challenge (Kain et al., 2013), and for this reason the research line focused in determining a way to include information relating to convection initiation into the numerical model is receiving greater interest in the research

community (Mecikalski et al., 2013). In general, radar instruments are located inland and thus, they are typically used to study extreme weather phenomena that takes place over land. Although they provide high temporal-space valuable information of the initiation and development of different kind of weather events, an obvious disadvantage is the fact they can not provide valuable information of weather events that are first initiated offshore and afterwards impacts populated coastal areas producing high socio-economic losses, such as tropical cyclones.

From the low oceanic and maritime observational coverage limitations obtained from meteorological radar instruments, the assimilation of satellite-derived observations emerged as a potential solution (Vukicevic et al., 2004, 2006; Polkinghorne et al., 2010; Polkinghorne and Vukicevic, 2011; Zupanski et al., 2011; Jones et al., 2013; Zhang et al., 2013). Current satellite instruments provide high spatial and temporal observations covering the whole globe, solving the radar problem over maritime

bodies. Assimilation of satellite observations are actually performed following two different approaches. The first approach consists in assimilating direct satellite infrared and microwave radiances via a radiative transfer model (RTM) (Vukicevic et al., 2004; McNally et al., 2006; Otkin, 2010; Zupanski et al., 2011). This approach has the main advantage that avoids uncertainties associated to various retrieval algorithms that differ from satellite to satellite (Derber and Wu, 1998; Errico, 2000) and generally performs best in clear-sky regions. However, the assimilation of cloudy radiances increases the uncertainties and

adds the complication of potential differences in cloud microphysics assumptions between the model and the RTM (Zupanski et al., 2011). Satellite radiances usually contain high correlated errors between different frequency bands (Bormann and Bauer, 2010), hampering its proper assimilation without introducing significant errors. Furthermore, radiance observations are frequently biased, which is particularly significant when convective-scales are considered, limiting considerably the assimilation and in consequence the forecast skill. Although recent bias correction methods have shown benefits in global data assimila-

tion configurations (e.g., Derber and Wu (1998); Fertig et al. (2009); Miyoshi et al. (2010)), the procedure for regional data assimilation is not well established. The second method used to assimilate satellite data is through the use of derived products, known as retrievals, such as profiles of geophysical quantities (T, q, CO, ...) or cloud water paths. Retrievals are easier to assimilate and interpret because they provide information that can be directly related with atmospheric variables, and its assimilation avoids the use of relatively complicated RTM. Although both methods are slightly different and contain different

types of errors associated, the overall information drawn from them has been found to be equivalent (Migliorini, 2012). From these reasons, in this study only satellite-derived products will be considered.





Over the last years, much efforts have been dedicated to improving the forecasting of Tropical Cyclones (TCs), which can cause flooding, heavy rain and strong winds resulting in numerous casualties and huge property damage. Despite the recent
improvements of numerical weather prediction (NWP) systems, the correct forecasting of TC track and intensity remains a big challenge (DeMaria et al., 2014). In addition, TCs life cycle occurs mainly over the ocean, where a lack of *in-situ* observations are present, and thus limiting our ability to determine potentially relevant aspects of the atmospheric state. Data assimilation techniques play a crucial role in improving the knowledge of initial conditions and the subsequent forecasts through the effective use of the available observations. Thus, assimilating special observations collected during field experiments (e.g., aircraft
or rawinsonde) and satellite-derived observations over the ocean is expected to produce a more accurate representation of the initial conditions and their respective forecasts of these extreme weather events. Among the wide variety of satellite-derived products, atmospheric motion vectors (AMVs) have been found to improve TC track forecasts in global numerical weather prediction systems (e.g., Le Marshall et al. (2008); Goerss (2009); Langland et al. (2009)). AMVs provide information of the local horizontal wind covering with great detail the mid- and upper tropospheric layers over the ocean. They are derived
from sequential satellite images by tracking the motion of targets including cirrus clouds, gradients in water vapour and small cumulus clouds (Velden et al., 1997). Much attention has been paid recently to the impact of assimilating AMV observations, as well as investigating the mitigation of the negative impacts of correlated data from remote sensing instruments. These studies highlight the impact of such observations on the TC track and structure, leaving the discussion about the intensity of the cyclone in the background (Houze Jr et al., 2006).


A rare type of Mediterranean cyclone has drawn the attention of the meteorological scientific community. These cyclones share some morphological characteristics with TCs, such as having a warm core, axisymmetry and cloud-free eye. In consequence, they are referred as medicanes, which is the acronym of Mediterranean hurricanes (Emanuel, 2005). These cyclones are relatively small in size and are associated to strong winds and heavy precipitations (Ernst and Matson, 1983; Rasmussen
and Zick, 1987; Lagouvardos et al., 1999; Fita et al., 2007), generating high impacts on exposed people and property assets (Jansa et al., 2001; Gómez et al., 2003; De Zolt et al., 2006). Although several studies have investigated this kind of phenomena (Jansa et al., 2014), physical mechanisms of medicanes are still poorly understood, and the numerical predictability of such events is considerably low. This is mainly due to the small size of these cyclones, the strong heat fluxes from the sea and the lack of *in-situ* observations present in the Mediterranean sea.


It is important to note that, although conventional and AMV observations have already been assimilated for some TCs events (e.g., Pu et al. (2008); Romine et al. (2013); Wu et al. (2014)), the conclusions obtained from these studies might not be suitable for medicane events taking place in the Mediterranean basin. The proper assimilation of observations in this area also represents a major challenge due to the lack of available *in-situ* observations and their location over complex topography,
seriously affecting the quality control process of such observations and thus their correct assimilation. Taking into account this challenge, one of the main objectives of this study is to find the most efficient way of using the available observations to obtain an accurate representation of the genesis and evolution of a medicane. Among the different available medicanes, the so





called Qendresa, which took place southern Sicily between 7-8 November 2014 (Carrió et al., 2017) and was poorly forecasted, was selected to perform this study. More precisely, the correct prediction of both the northward loop trajectory followed by Qendresa and its intensification still remain a major challenge for most current numerical weather models. Results obtained from high-resolution numerical sensitivity experiments indicated the relevant role of the upper-level PV effect on its cyclogenesis during early stages of the event (Carrió et al., 2017), helping to set up a favorable environment for deep convection, intense latent heat release and thus, the medicane development. Therefore, with the aim of improving the representation of the upper-level dynamics of Qendresa and consequently its predictability, the assimilation of AMV observations is conducted in this study.

In overall, the present study is focus for the first time, on the improvement of the predictability of the Qendresa medicane through the assimilation of *in-situ* conventional and high temporal and spatial resolution satellite-derived observations using the advanced EnKF data assimilation technique. This study is structured as follows. In section 2 a brief description of the Qendresa medicane is presented. Section 3 provides information about the numerical tools and the experimental configuration used for the simulations that are planned to be performed. Results from the assimilation of different observations and their impact in the short-range forecast are provided in section 4. Finally, conclusions and further work is presented in section 5.

## 2  Description of the 7 November 2014 Medicane Qendresa

During 7 November 2014 an intense small cyclonic system with tropical-like features (medicane) affected the Western and Central Mediterranean basin. This small cyclonic system initiated in the Sicily channel during the first hours of 7 November and evolved east-north-eastward, reaching its maximum of intensity over Malta island, where it was categorized as a medicane. At this moment, a small well-defined spiral-to-circular cloud shape showing a clear eye was visible through *Meteosat Second Generation* imagery (Fig. 1). In addition, intermittent deep convection around the eyer was also observed. In the following hours, the medicane continued progressing north-eastward towards the eastern coast of Sicily (Catania), where it started its dissipation phase when it made landfall.

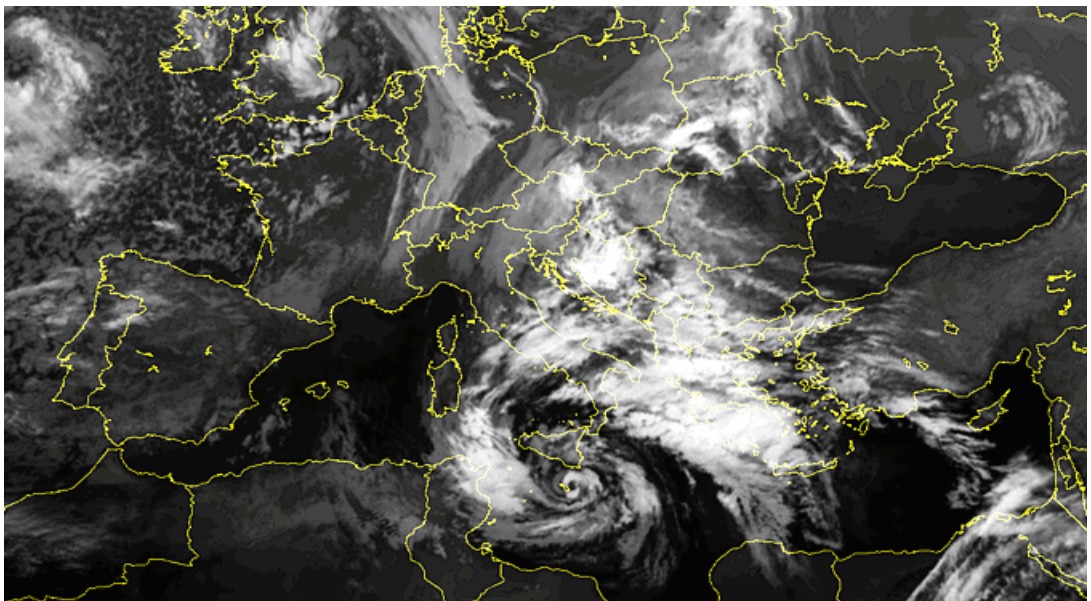

**Figure 1.** IR-MSG satellite imagery at 1800 UTC on 7 November 2014.

This episode was influenced by the upper-level synoptic situation that was mainly dominated by an intense trough associated with a PV streamer extending from northern Europe to southern Algeria (Africa). At the surface, general cyclonic circulation was present over the Western Mediterranean influenced by the North Atlantic Subtropical High system and a high-pressure system over the Eastern Mediterranean. During 6 November, the upper-level trough became negatively tilted and the PV streamer eventually disconnected from the northern nucleus, evolving into an intense upper-level cut-off during the late hours of 7 November. A more detailed description of the synoptic situation associated to this medicane event and the physical mechanisms involved in the genesis and posterior evolution of this medicane through high-resolution numerical sensitivity experiments can be found in Carrió et al. (2017). Overall, results obtained from these sensitivity experiments indicated the relevant role of the upper-level PV effect on the medicane development.

The present study aims to highlight the main challenges in accurately predicting this type of small-scale intense cyclones in terms of both trajectory and intensity features, in addition to shedding light on the limitations exposed by sophisticated data assimilation techniques, such as the EnKF. It is hypothesized that the low predictability of Qendresa is likely associated with the difficulty of current numerical weather models to depict accurately the atmospheric state (i.e., initial conditions) of the cyclone over the sea (southern Sicily), due to the lack of *in-situ* observations present. To assess the potential of the EnKF in improving the predictability of this extreme weather event, we will compare the trajectory and intensity features obtained by the assimilation of different available weather instruments. Due to the lack of *in-situ* observations present over the maritime region where the medicane took place, infrared imagery was used as *a proxy* to estimate the track followed by Qendresa (Fig. 2). It is observed how the cyclone initiated over the Pantelleria region and during the subsequent hours it moved eastwards,

becoming a tropical-like cyclone, and sweeping across Malta, where it was reflected northeastwards, reaching the offshore of Catania. Finally, when the cyclone started its dissipation phase, the cyclone suffered an abrupt cyclonic counter-clock deviation, towards the eastern coast of Sicily, when it made landfall. This loop-like trajectory is the main feature that current numerical

models do not accurately predict mainly due to a miss representation of the upper-level dynamics. Therefore, the assimilation of AMVs is expected to improve the prediction of such curvature in the trajectory of the cyclone, in addition to also improving the prediction of its intensity.

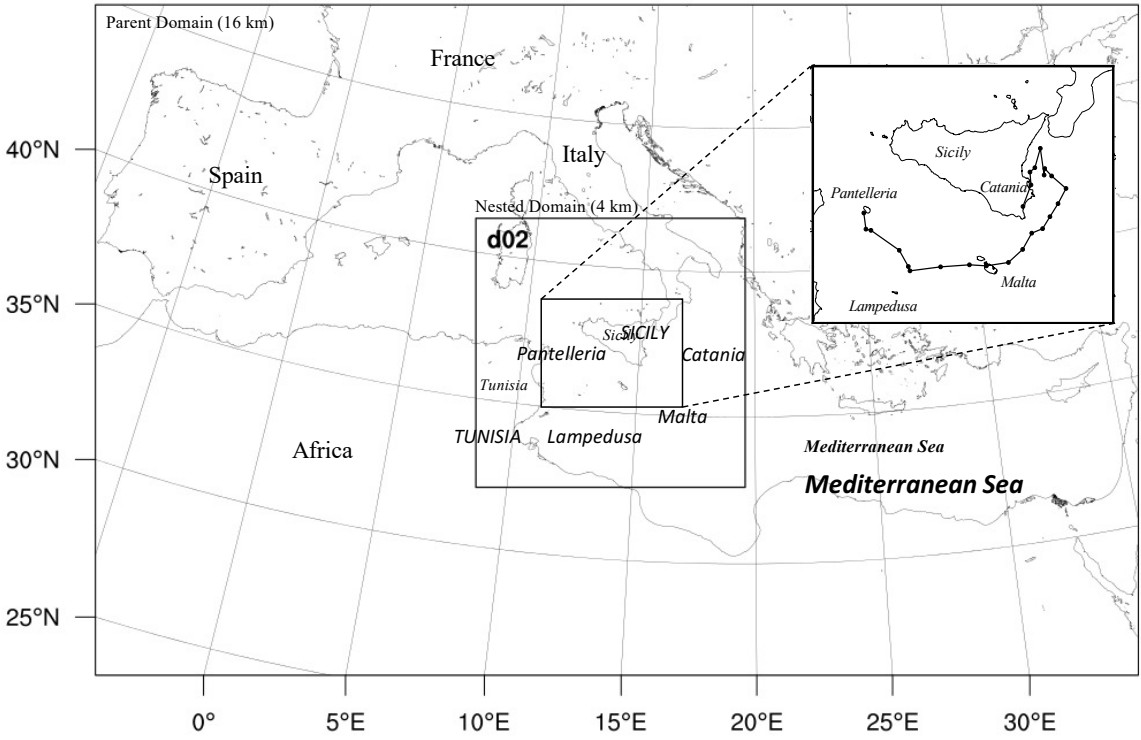

**Figure 2.** Numerical domain used for the multi-scale simulations performed in this study. The top-right embedded diagram shows the observed track of the medicane viewed from infrared satellite imagery.


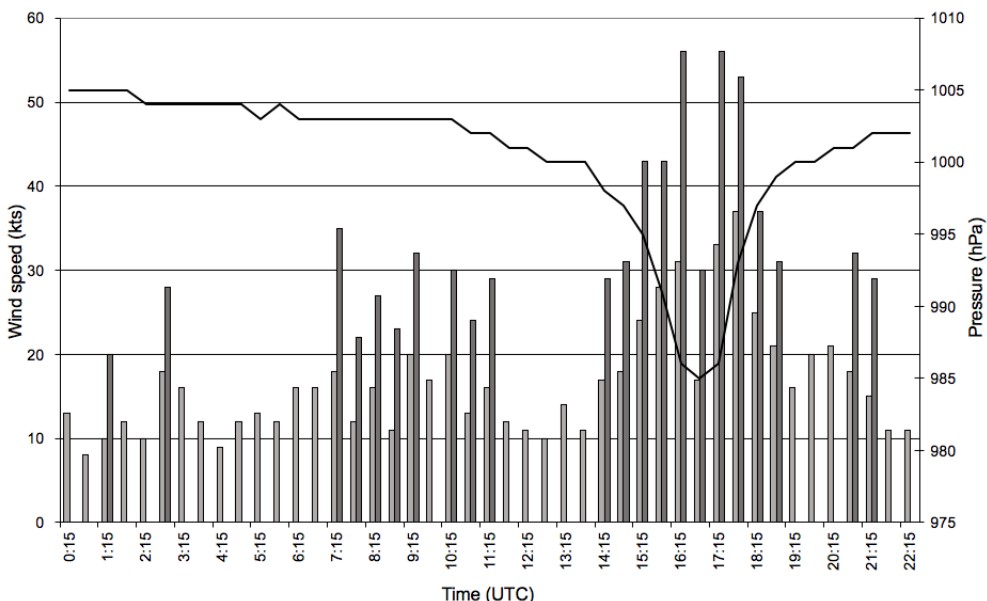

**Figure 3.** Surface pressure (hPa, solid line), sustained wind and gusts ( m $s^{-1}$, light and dark grey bars respectively) form METAR stations, at (a) Pantelleria, (b) Lampedusa, (c) Malta and (d) Catania. Indicated times in UTC of 7 November 2014.

In terms of the intensification of the medicane, and taking into account the scarce number of observations over the sea, METAR reports from land locations close to the cyclone path (i.e., Pantelleria, Lampedusa, Malta and Catanian coast for geographical references) were used to provide direct measurements of the small cyclone (Fig. 3). In particular, during the afternoon of 7 November the center of the cyclone crossed right over Luqa airport (Malta), so the pressure registered by the station can be treated as a good representation of the surface pressure section across the cyclone at that time. The barometer recorded a pressure drop of nearly 20 hPa in 6 h and a minimum pressure value of 985 hPa at 1645 UTC. In addition, wind values registered in Malta also provides valuable information about the shape and symmetry of the cyclone. In particular, intense wind speeds with gusts exceeding 42.7 m $s^{-1}$ that precedes and follows a relatively calmed period around 1645 UTC provides a clear signature of the cyclone's eye structure (Fig. 3).

## 3 Numerical Tools and Experimental Setup

### 3.1 Numerical Weather Model

The numerical simulations presented in this study are performed using the Version 3.7 of the Advanced Weather Research and Forecasting Model (WRF-ARW, Skamarock et al. (2008)). These simulations used a multi-scale ensemble system based on two one-way nested domains to better account for meso- and storm-scale processes involved in the genesis and evolution of Qendresa (Fig. 2). The parent domain is centered over the Central part of the Mediterranean Sea, covering most of the European





region and the northern part of Africa, with horizontal grid resolution of 16 km and 51 terrain-following $\eta$ levels up to 50 hPa (245x245x51). The nested domain is centered over Sicily (southern Italy) with a grid resolution of 4 km (253x253x51). The numerical experiments are performed using a 36-member ensemble, which was designed as in (Carrió and Homar, 2016), using initial and boundary conditions from the EPS-ECMWF, which has an horizontal and vertical spectral triangular truncation of T639L62 ($\sim$ 31 km horizontal grid resolution).


To address the uncertainties in the numerical model different combinations of physics schemes were used among its members (Table 1). In particular, the physics variability includes three planetary boundary layer (PBL) schemes [Dudhia (Dudhia, 1989); Mellor-Yamada-Janjic (Janjic, 1990) and Mellor-Yamada-Nakanishi-Niino level 2.5 (Nakanishi and Niino, 2006)], two short- and longwave radiation schemes [Dudhia (Dudhia, 1989) and RRTMG (Iacono et al., 2008)], and three cumulus param-

eterizations schemes [Kain-Fritcsh (Kain, 2004); Tiedke (Tiedtke, 1989) and Grell-Freitas (Grell and Freitas, 2013)]. Common to all the ensemble members we have the Thompson microphysics (Thompson et al., 2004). Note that the above mentioned parameterization schemes are identical for the parent and inner numerical domains.

### 3.2  Observations

In order to improve the initial conditions of Qendresa over the sea and in consequence, its forecast towards populated areas, two

different observational sources of information were assimilated: the *in-situ* conventional and the satellite-derived observations. Conventional *in-situ* observations were obtained from the global *Meteorological Assimilation Data Ingest System* (MADIS), where meteorological variables such as temperature, humidity, pressure and both wind speed and direction are collected from rawinsondes, buoys, ship reports, aircrafts (ACARs) and also from meteorological aerodrome reports (METARs), among others. One of the main advantages of using MADIS observations is the fact that they are quality controlled through spatial

and temporal consistency checks[1]. For this particular study, only observations from buoys, METARs and rawinsondes have been considered, providing data every hour and covering the entire Mediterranean region. However, it is noteworthy that the maritime area where the medicane took place remain poorly observed by *in-situ* instruments (Fig. 4).

---

[1]Details on the quality control procedures can be found online at http://madis.noaa.gov/madis_qc.html.



**Table 1.** Multi-physic parameterizations used on the WRF ensemble system presented on this study. Here PBL, SW and LW stand for planetary boundary layer, shortwave and longwave respectively.

| Multiphysic Configuration | | | | | |
|---|---|---|---|---|---|
| Ensemble Members | Microphysics | Cumulus | PBL | Land Surface | SW/RW radiation |
| 1 | Thompson | KF | YSU | Noah | Dudhia |
| 2 | | KF | YSU | | RRTMG |
| 3 | | KF | MYJ | | Dudhia |
| 4 | | KF | MYJ | | RRTMG |
| 5 | | KF | MYNN2 | | Dudhia |
| 6 | | KF | MYNN2 | | RRTMG |
| 7 | Thompson | GF | YSU | Noah | Dudhia |
| 8 | | GF | YSU | | RRTMG |
| 9 | | GF | MYJ | | Dudhia |
| 10 | | GF | MYJ | | RRTMG |
| 11 | | GF | MYNN2 | | Dudhia |
| 12 | | GF | MYNN2 | | RRTMG |
| 13 | Thompson | Tiedke | YSU | Noah | Dudhia |
| 14 | | Tiedke | YSU | | RRTMG |
| 15 | | Tiedke | MYJ | | Dudhia |
| 16 | | Tiedke | MYJ | | RRTMG |
| 17 | | Tiedke | MYNN2 | | Dudhia |
| 18 | | KF | MYNN2 | | RRTMG |
| 19 | Thompson | KF | YSU | Noah | Dudhia |
| 20 | | KF | YSU | | RRTMG |
| 21 | | KF | MYJ | | Dudhia |
| 22 | | KF | MYJ | | RRTMG |
| 23 | | KF | MYNN2 | | Dudhia |
| 24 | | KF | MYNN2 | | RRTMG |
| 25 | Thompson | GF | YSU | Noah | Dudhia |
| 26 | | GF | YSU | | RRTMG |
| 27 | | GF | MYJ | | Dudhia |
| 28 | | GF | MYJ | | RRTMG |
| 29 | | GF | MYNN2 | | Dudhia |
| 30 | | GF | MYNN2 | | RRTMG |
| 31 | Thompson | Tiedke | YSU | Noah | Dudhia |
| 32 | | Tiedke | YSU | | RRTMG |
| 33 | | Tiedke | MYJ | | Dudhia |
| 34 | | Tiedke | MYJ | | RRTMG |
| 35 | | Tiedke | MYNN2 | | Dudhia |
| 36 | | Tiedke | MYNN2 | | RRTMG |


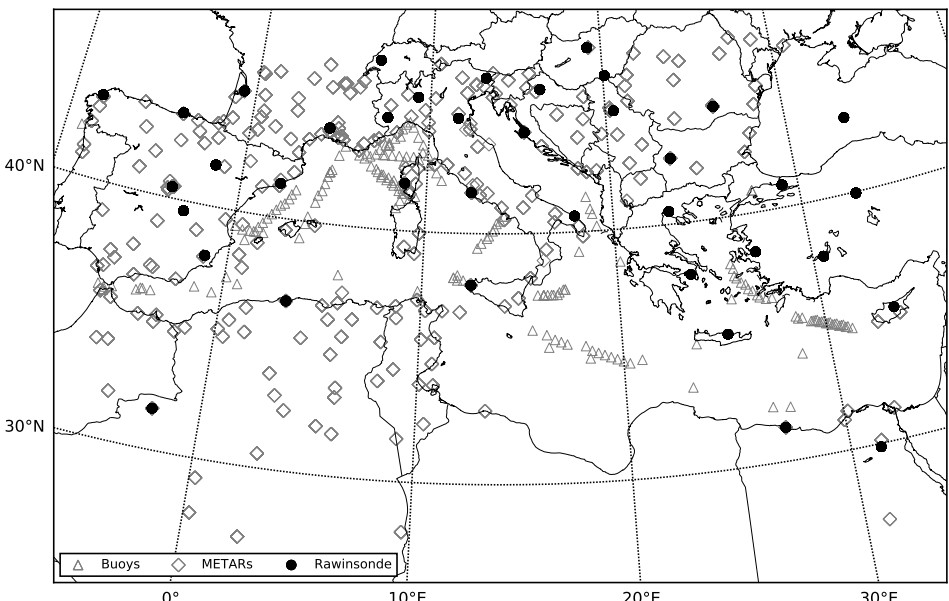

**Figure 4.** Spatial distribution of *in-situ* conventional observations from MADIS database (buoys, METARs and rawinsondes) assimilated every hour between 12 UTC on 06 November and 00 UTC on 07 November 2014.

With the main aim of improving the representation of the state of the atmosphere over those poorly observed regions over the sea, a cloud based satellite-derived product known as *Atmospheric Motion Vectors* (AMVs; Schmetz et al. (1993)), which

provides high spatial and temporal resolution information about the local horizontal winds (both speed and direction values) at different vertical pressure levels was used. Taking into account that the upper-level dynamics played a key role in the genesis and development of this medicane (see Carrió et al. (2017)) it is expected that the assimilation of such observations will contributes to significantly enhance the predictability of this medicane, improving both its trajectory and intensity. Although the potential effect of assimilating this kind of observations has already been assessed for TCs (Pu et al., 2008; Goerss, 2009;

Langland et al., 2009; Romine et al., 2013; Wu et al., 2015), it has not been assessed over short life-span smaller cyclones, such as medicanes, which are less predictable than tropical cyclones. Indeed, as far as the author is aware at the time of writing this study, the present study assesses for the first time the potential of assimilating AMVs to improve the predictability of a medicane.

Although the assimilation of AMVs is promising to improve the short-range forecast of Qendresa, its 1 hour temporal resolution availability could be a disadvantage. Instead, a more frequent satellite-derived wind product from EUMETSAT database, referred to as *Rapid-Scan Atmospheric Motion Vectors*[2] (RSAMVs), which provides data every 20-min, is used (Fig. 5). In the present study, RSAMVs observations at all heights below the tropopause are obtained from 5 different spectral channels (water

---

[2]This product is obtained using the *Spinning Enhanced Visible and Infrared Imager* (SEVIRI) instrument on board the *Meteosat Second Generation* (MSG) satellite, which has a scanning frequency as low as 5 minutes. The final product is obtained averaging 4 consecutive images.





vapour 6.2 $\mu m$, water vapour 7.3 $\mu m$, infrared 10.8 $\mu m$, visual 0.8 $\mu m$ and high resolution visual channel), all combined into
one product. Notice that the spatial coverage of this product is limited along the latitudes from 35ºN to 90ºN, and for this reason
no wind information is depicted in the bottom part of the numerical domain showed in Fig. 5.

When dealing with RSAMV observations it is important to note that the different spectral channels do not identify exclusively different layers of the atmosphere, they overlap. In other words, one spectral channel can identify the same wind
observation that another channel can identify. However, is not common that both different channels provides precisely the
same value of such observation. For this reason, to avoid possible inconsistencies in the DA process due to the assimilation
of different observation values from different spectral channels, only the wind observations from infrared 10.8 $\mu m$ are assimilated, which provides wind information for the entire troposphere. Before assimilating these observations, a quality control
(QC) check was also performed to reject non-physical and/or outliers observations that could deteriorate the quality of the
analysis and its successive forecast. Specifically, in this study four main QC checks were performed to all the conventional and
RSAMV observations. In the first QC check, surface observations were rejected if the model terrain and observational height
amounts differ significantly from each other. This is applied to avoid inconsistencies between the physical representation of
topography in the numerical model and the actual measurements. In this study, a threshold of 500 m was used. The second
QC check consists on reducing instabilities or noise generated by the assimilation of observations near the domain boundaries
(see Romine et al. (2013)). This procedure is performed excluding observations located within 5 grid lengths from the lateral boundaries of the numerical model. The third QC check is applied to reject outlier observations, such observations with
non-physical values or those which would result in unacceptably large increments in the analysis after their assimilation, that
could destabilize the numerical model. This method is based on the square difference between the prior ensemble mean and the
observation values. In this study, if the squared difference exceeds 3 times the sum of the prior and the observation error vari-
ances, then the observation is rejected. Finally, to minimize the unwanted effect of having spatial correlated errors associated to
high density observations, such as RSAMVs, a *superobbing* technique consisting in reducing the data density through spatially
averaging the observations within a predefined prism is applied (e.g., Pu et al. (2008); Romine et al. (2013); Wu et al. (2013);
Honda et al. (2018)). In this study, RSAMV observations were thinned using a prism with horizontal dimensions of (128 x 128)
$km^2$ and 25 hPa vertical extent. It is important to note that this prism dimension values were determined after several sensitiv-
ity simulations, in which they provided the most accurate analysis of the environment conditions preceding the medicane event.




**Figure 5.** Raw EUMETSAT's RSAMV observations depicted at different vertical levels by different channels at 1200 UTC on 07 November 2014 over the Mediterranean region: a) high resolution visual channel 0.75 $\mu m$, b) Water vapor 6.2 $\mu m$ and c) infrared 10.8 $\mu m$. RSAMV final product (combination of all spectral channels) is also depicted in figure panel d). Wind information is only valid at the center of the wind vectors.

In order to assimilate these two types of observations (i.e., conventional and satellite-derived) it is crucial to assign the corresponding observational error value, which in this case is measured by the standard deviation. Following similar studies (e.g., Romine et al. (2013); Yussouf et al. (2015); Carrió and Homar (2016)), the observational error values used here for the conventional observations are: 0.75 K for the temperature, 0.75 K for the dew point temperature, 0.75 $m\ s^{-1}$ for the wind





speed and 0.75 hPa for the pressure. Regarding the observational error associated with RSAMV, and after several sensitivity tests tuning this parameter, a standard deviation error of 1.4 $m\ s^{-1}$ was used, which produced the best analysis.

### 3.3 EnKF Data Assimilation Configuration

The ingestion of the above mentioned observations within the WRF numerical weather model is expected to improve the
estimation of the precondition thermodynamic environment that leads to the formation of the Qendresa medicane and in consequence, enhance its short-range track and intensity forecasts. To this purpose, the data assimilation scheme used for this study is the parallel version of the *Ensemble Adjustment Kalman Filter* (EAKF; Anderson (2001)) from the Trunk release branch (revision 9240) of the *Data Assimilation Research Testbed* software system (DART; Anderson and Collins (2007); Anderson et al. (2009)).


In particular, the conventional and satellite-derived observations were assimilated with hourly and 20-min frequency respectively, using a data assimilation window from 12 UTC 06 November to 00 UTC 07 November 2014. In each data assimilation cycle the EAKF algorithm updates a set of model prognostic variables and also some diagnostic fields. In this study the EAKF updates the three dimensional wind fields, the potential temperature perturbation, the geopotential perturbation and the dry air
surface pressure perturbation, as well as the water vapor and the following hydrometeor fields: mixing ratio of cloud, rain, ice, snow, graupel and the number of concentration of rain and ice. Regarding the diagnostic fields the EAKF updates the 10-m wind fields, 2-m temperature, 2-m moisture and surface pressure.

The computational cost of implementing DA schemes in real applications such this study forces to use a moderate ensemble
size (i.e., 20-50 members). However, covariance estimates obtained from this modest ensemble produces spurious correlations that severely deteriorates the analysis from the data assimilation process (Hacker et al., 2007). To reduce the negative effect of sampling errors a localization technique (Houtekamer and Mitchell, 1998) based on a distance weighting function that tends to zero as the distance from the observation location increases is used (Sobash and Stensrud, 2013). In this study a Gaussian localization function referred as the fifth-order piece-wise rational function (Gaspari and Cohn, 1999) is used. For the horizontal
and vertical localizations, a half-radius of 510 km and 3 km is used, respectively.

Associated with the use of a moderate ensemble there exists an important issue related with the collapse of the ensemble spread after each analysis cycle (Anderson and Anderson, 1999), which will also deteriorate the analysis from the DA process. To avoid having such an under-dispersive system, an adaptative spatially and temporally varying inflation technique is applied
to the prior ensemble state vector (i.e., before the assimilation step) in each assimilation cycle (Anderson and Collins, 2007; Anderson et al., 2009). This technique requires two parameters: the mean of the initial inflation factor and its standard deviation. According to previous studies (e.g., Carrió et al. (2019)) a mean initial inflation value of 1.0 with 0.6 of standard deviation was used here.



## 3.4 Experimental Setup

To quantitatively assess the potential impact of assimilating conventional, and specially RSAMV observations, in the forecast skill of Qendresa, the following four different data assimilation simulations were performed (Fig. 6).

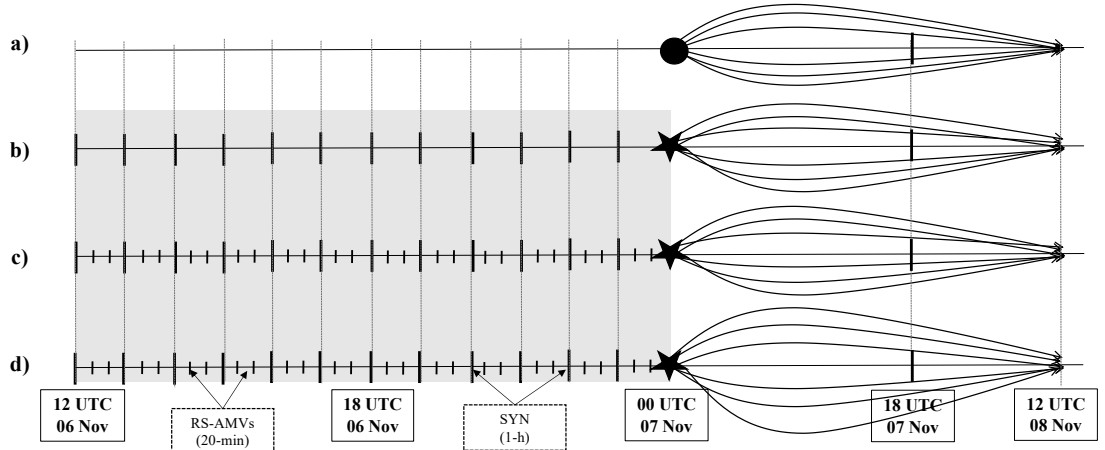

**Figure 6.** Schematic representation of the different numerical experiments designs performed in this study: (a) NODA, (b) SYN, (c) RSAMV and (d) CNTRL DA experiments. The shaded area corresponds to the assimilation windows between 12 UTC 06 November to 00 UTC 07 November where observations are assimilated. Black dot represent the initiation time of the forecast for the NODA experiment. Black stars represent the final analysis obtained from the different DA experiments and from which the subsequent free forecast is initiated.

### 3.4.1 NODA Experiment

With the main objective of assessing the value of assimilating different types of observations, it is required to have an analogous numerical forecast but without assimilating observations (NODA). In this way, the effects of assimilating different types of observations can be assessed by a direct comparison between NODA and the other DA experiments. NODA experiment is simply a direct downscaling from ECMWF-EPS at 00 UTC 07 November to 12 UTC 08 November 2014 (Fig. 6a). It is important to highlight that the choice of starting the NODA experiment at 00 UTC 07 November, instead of starting at 12 UTC 06 November, was made intentionally to extract general conclusions applicable to an operational framework. In other words, we are interested in comparing the DA experiments using observation until 00 UTC 7 November that could operationally be performed during the first hours of 7 November, with the most accurate experiment without assimilating observations that could also be run at the same time. Therefore, the forecast differences between each DA experiment and NODA will directly be attributed to the impact of the observations and also to the DA configuration used in each DA experiment.



### 3.4.2 SYN Experiment

SYN is intended to improve synoptic and meso scale environment through the assimilation of *in-situ* conventional observations. In particular, such observations are METARs, rawinsondes and buoys. To achieve this objective, the following experiment consists in two main stages: (i) the analysis update obtained by the data assimilation procedure and (ii) the free ensemble forecast initialized from such analysis. The analysis update is designed to hourly assimilate conventional observations within an assimilation window from 12 UTC 06 November to 00 UTC 07 November 2014. At the end of this process, a new estimation of the atmospheric stat due to the assimilation of the above mentioned conventional observations is obtained. Then, a free 36 hour forecast is launched using the ensemble analysis obtained at 00 UTC 07 November as initial conditions (Fig. 6b).

### 3.4.3 RSAMV Experiment

This experiment is analogous to SYN, with the main difference that only RSAMV observations from the EUMETSAT database are assimilated. However, in this case the updated frequency for assimilating these observations has increased to 20-min instead to the hourly data assimilation frequency used in SYN (Fig. 6c). It is expected that the assimilation of RSAMVs will contribute to enhance the numerical representation of the atmospheric flow, specially the upper-level dynamics, which have been shown key during the cyclogenesis of Qendresa. The assimilation of RSAMV has already been shown to contribute to the improvement of the intensity and track forecasts of TCs. However, it is important to highlight that these studies typically assimilate such observations once TCs are fully developed (i.e., mature phase). In this case we are interested to study the impact of assimilating these observations on the pre-convective environment that leads to the initiation and posterior evolution of this medicane.

### 3.4.4 CNTRL Experiment

The CNTRL experiment is designed with the main objective of assessing the potential effect of assimilating both *in-situ* conventional and RSAMV observations to improve the track and the intensification forecasts of Qendresa (Fig. 6d). Specifically, conventional and RSAMV observations are both assimilated on the hour (i.e., 1200 UTC, 1300 UTC, ...) and only RSAMV observations are assimilated every 20-min (i.e., 1200 UTC, 1220 UTC, 1240 UTC, 1300 UTC,...). Similar to the above-mentioned experiments, after obtaining the last analysis at the end of the assimilation window (00 UTC 07 November), a 36-h free ensemble forecast is launched from such analysis. This experiment has the advantage of taking benefit of the RSAMV observations to improve the representation of the upper-level atmospheric circulation and also accounting for the surface conventional observations improving the analysis at low levels and mainly over the sea surface, where analysis errors are larger than over land.





## 4 Results

### 4.1 Observation-space diagnostics

To quantitatively assess the data assimilation performance during the 12-h data assimilation window, the following widely-
used observation-space diagnostics (Yussouf et al., 2013; Wheatley et al., 2015; Carrió et al., 2019) are computed using the
background and EnKF analysis model states mapped to the observation locations: (i) the root mean square error innovation
(rmsi), (ii) the bias (model-obs), (iii) the total ensemble spread (TS; Dowell and Wicker (2009)) and finally (iv) the consistency
ratio (CR; Dowell et al. (2004)). These diagnostics are computed for the entire set of assimilated type of observations (i.e.,
METARs, buoys, rawinsondes and RSAMVs) before and after each hourly data assimilation cycle. However, for the sake of
brevity only results from METARs diagnostics are shown here. More details of these diagnostics can be found in Carrió et al.
(2022).

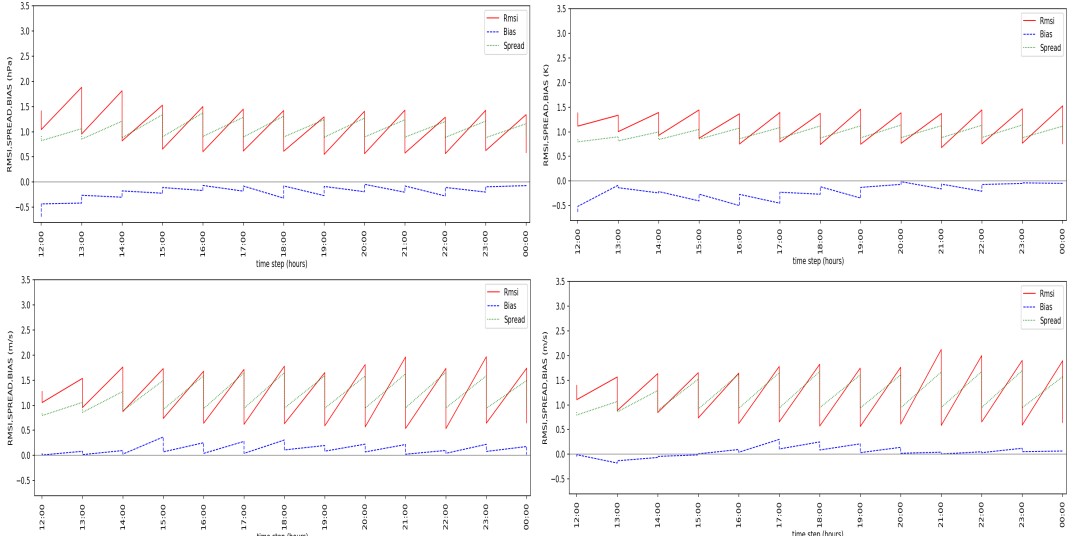

**Figure 7.** Sawtooth plots depicting the evolution of the RMSE, Bias and total spread during the assimilation window for the METAR (a)
altimeter and (b) temperature observations, and for the RSMAV (c) U-wind, (d) V-wind observations.

Results from these diagnostics clearly show the positive effect of the assimilation process, reducing significantly the rmsi
after each data assimilation cycle for all kind of observations (Fig. 7). In agreement with these results, the bias score is also
showing a significant decrease with time for the different observation types. For METAR observations, the bias values indicate
that the model under-predicts the observed variables (Fig. 7 (a,b)). On the contrary, bias associated with RSAMVs indicates
that, generally, the model over-predicts the observed wind observations (Fig. 7 (c,d)). The evolution of the total spread (TS)
shows values tending to an approximated value of 0.9, indicating that the data assimilation system performance is stable. In
addition, the evolution of TS is similar to the RMSE evolution, resulting in consistency ratio (CR) values near 1.0 (Fig. 8).
Recall that a consistency ratio of 1.0 indicates that the prior spread of the ensemble is a good approximation of the forecast
325 error taking into account the observational error assigned. For this case, CR values gets closer to 1.0 at the end of the data assimilation window, confirming that our prior ensemble is a fair sample from the truth distribution.

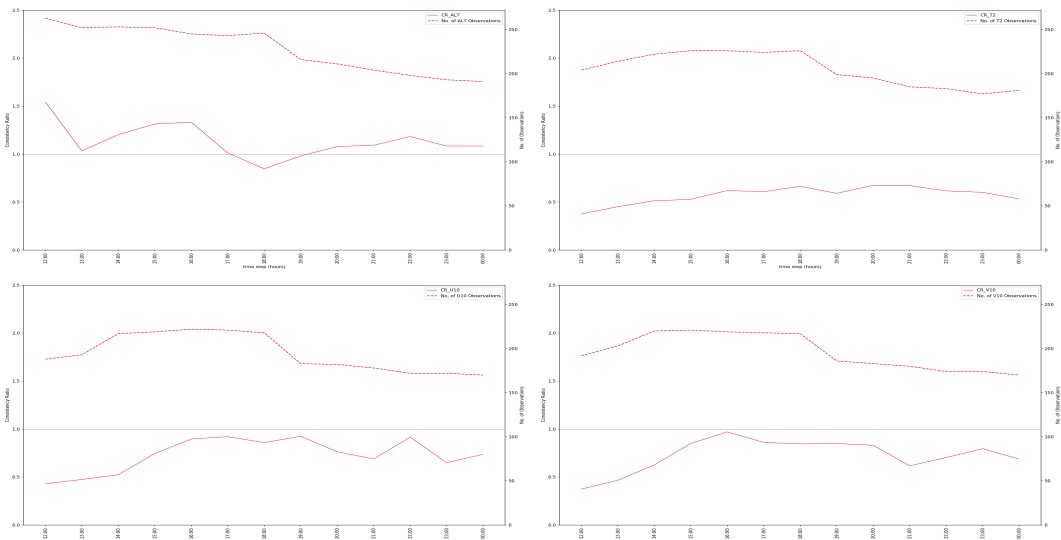

**Figure 8.** Evolution of the CR during the assimilation window for the METAR (a) altimeter and (b) temperature observations, and for the RSAMV (c) U-wind and (d) V-wind observations. The number of observations assimilated at each data assimilation cycle is depicted by the dotted line.

To further evaluate the performance of the data assimilation system, an additional diagnostic was used in this study. This diagnostic computes the squared correlation coefficient $R^2$ between the observations and model values. In particular, $R^2$ is computed using the prior and posterior states to show the effect of the data assimilation system on the model state (Fig. 9). In

330 this case, Fig. 9 clearly shows the benefits of assimilating the RSAMVs obtained at the end of the data assimilation window at 00 UTC 07 November: before the assimilation of RSAMVs the prior state differs significantly from the observations, as observed by the off-diagonal distribution of grey dots. However, after the assimilation of RSAMVs, the posterior state matches closely the observations, as the blue points indicated being distributed over the diagonal. This effect is quantitatively measured by the increment of $R^2$ from 0.874 to 0.996.

335

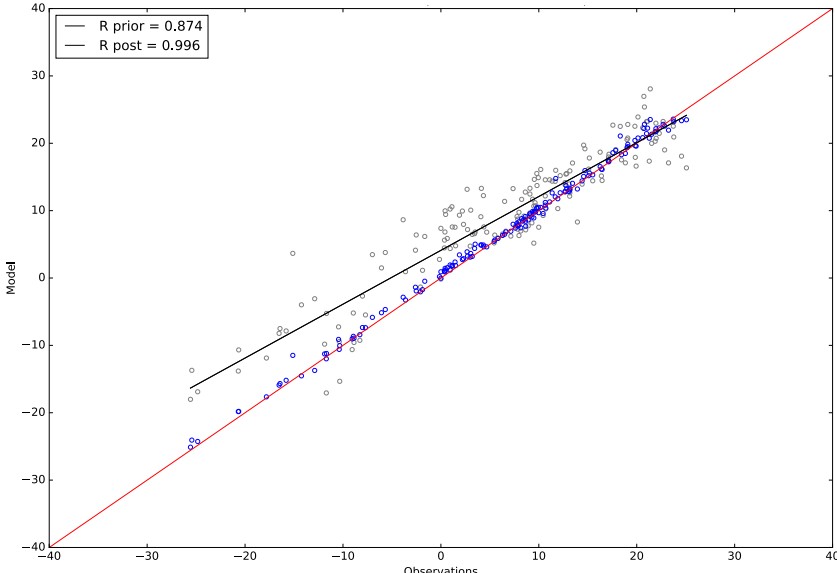

**Figure 9.** Squared correlation coefficient computed before (grey dots) and after (blue dots) the assimilation of pressure data from maritime buoys, referred as *prior* and *posterior*, at 00 UTC 07 November. The black line represents the prior linear regression line and the red line represents a perfect linear regression line.

To understand the impact of the assimilation of different types of observations along the data assimilation window, the evolution of the squared correlation score for each type of observations assimilated (i.e., METARs, maritime (buoys), U-component and V-component of the wind from RSAMVs) is computed. Results show high squared correlation values for almost the assimilation cycles, indicating a good correspondence between the posterior model state and the observations. In the case of the
340 U-component of the three dimensional wind, a lower correspondence between the model and the observations during the first assimilation cycles is observed. However, as new RSAMVs observations are assimilated in the following steps, the squared correlation increases. At the end of the assimilation window, the squared correlation values for the wind U-component become practically the same as the V-component values.


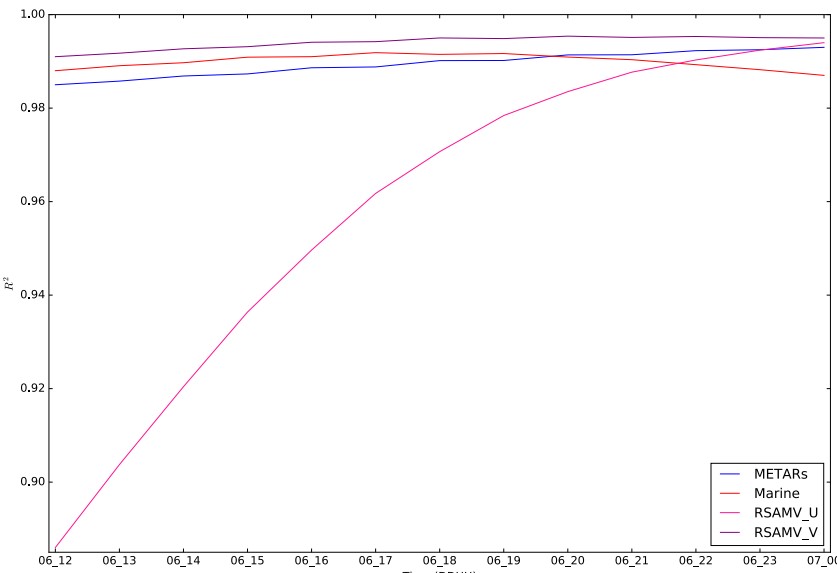

**Figure 10.** Evolution of the squared correlation for different observation types assimilated during the entire assimilation window.

## 4.2    Model-space diagnostics

Once we obtained the EnKF analyses at 00 UTC 07 November, we started a 36-h free ensemble forecast for each of the numerical experiments (i.e., NODA, SYN, RSAMV and CNTRL). Due to the inherent difficulty to accurately predict the observed trajectory and intensity of this medicane event (Pytharoulis et al., 2018), the potential impact of assimilating the above-mentioned observations to simulate the observed trajectory of the medicane and its intensification is investigated.

### 4.2.1    DA impact on Qendresa's Track forecast

Among the entire set of analysis obtained from the different numerical experiments, NODA was initially expected to produce an accurate representation of the medicane trajectory. However, forecast results show that most ensemble members do not properly simulate the observed trajectory, particularly the loop-ending on the eastern coast of Sicily, which is produced when the cyclone made landfall and its dissipation phase begin (Fig. 11a). In general, most ensemble members show a rapid northward evolution towards the central part of Sicily and then abruptly change direction towards the south-east. For the SYN experiment, which assimilates conventional observation, the cyclone trajectory follow a 'U' shape (i.e., first moving towards the Southeast, then moving to the east and finally moving towards the north-east) similar to the trajectory observed over satellite imagery (Fig. 11b). Although the shape of the trajectory agrees with observations, the location is not accurate. In general, trajectories are shifted towards the east. In the case of the RSAMV experiment, in which only wind observations from satellite are assimilated, a similar behavior is detected, but now more diversity among ensemble members is observed (Fig. 11c).

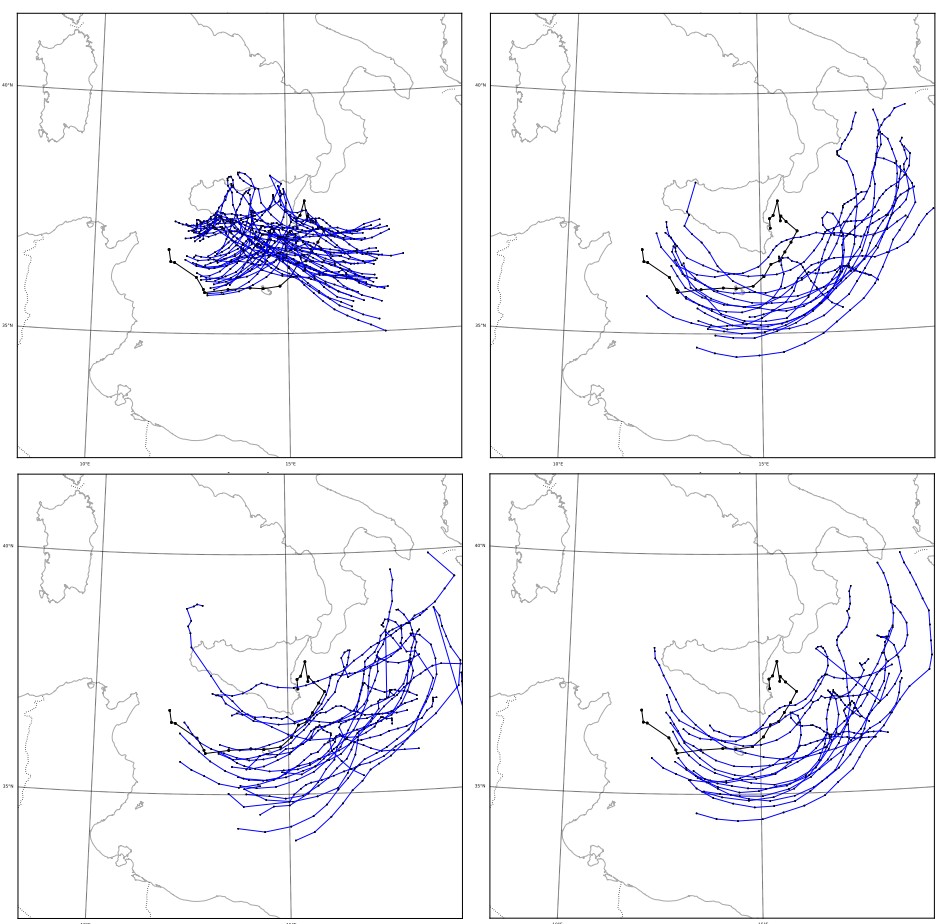

**Figure 11.** Trajectory of cyclones generated by each ensemble member (blue solid lines) by the (a) NODA, (b) SYN, (c) RSAMV and (d) CNTRL experiments from 11 UTC 07 November to 12 UTC 08 November 2014. Black solid lines depict the trajectory of the medicane observed from satellite imagery.

Some ensemble members simulate intense cyclones that follow a trajectory that completely differ from observations. Finally, the CNTRL experiment, which assimilates conventional and satellite wind observations, shows cyclone trajectories resembling the SYN experiment without significant discrepancies (Fig. 11d). In the case of the data assimilation experiments (i.e., SYN,
RSAMV, and CNTRL), no cyclone signature was identifiable during the first hours of forecast. It was not until 11 UTC 7 November, when small vortex circulations appeared. For this reason, the trajectories depicted in Fig. 11 corresponds to the period from 11 UTC 7 November to 12 UTC 8 November. It is also noteworthy that for the data assimilation experiments not all ensemble members depict cyclonic circulations. In fact, for the SYN experiment only 17/36 ensemble members generate a small-scale isolated cyclone, while in the RSAMV, a reduced number of members simulate cyclones (16/36), and finally in the
CNTRL experiment, this number is increased to 21/36. These results reveal the positive effect of assimilating both conventional





and RSAMVs to improve the predictability of this medicane event.

Looking the ensemble of cyclone's trajectories for each experiment we can state that some of them depict the evolution of the cyclone significantly different to the observed one, but some of them also depict trajectories resembling the observations.

Taking into account this, we have also represented the best track simulated by the different experiments in comparison with the trajectory observed by satellite imagery (Fig. 12). Among these results, only the CNTRL experiment is able to depict the curvature (i.e., loop) of the cyclone trajectory in a similar way that is observed. However, the entire track of the cyclone has a notable southward bias.

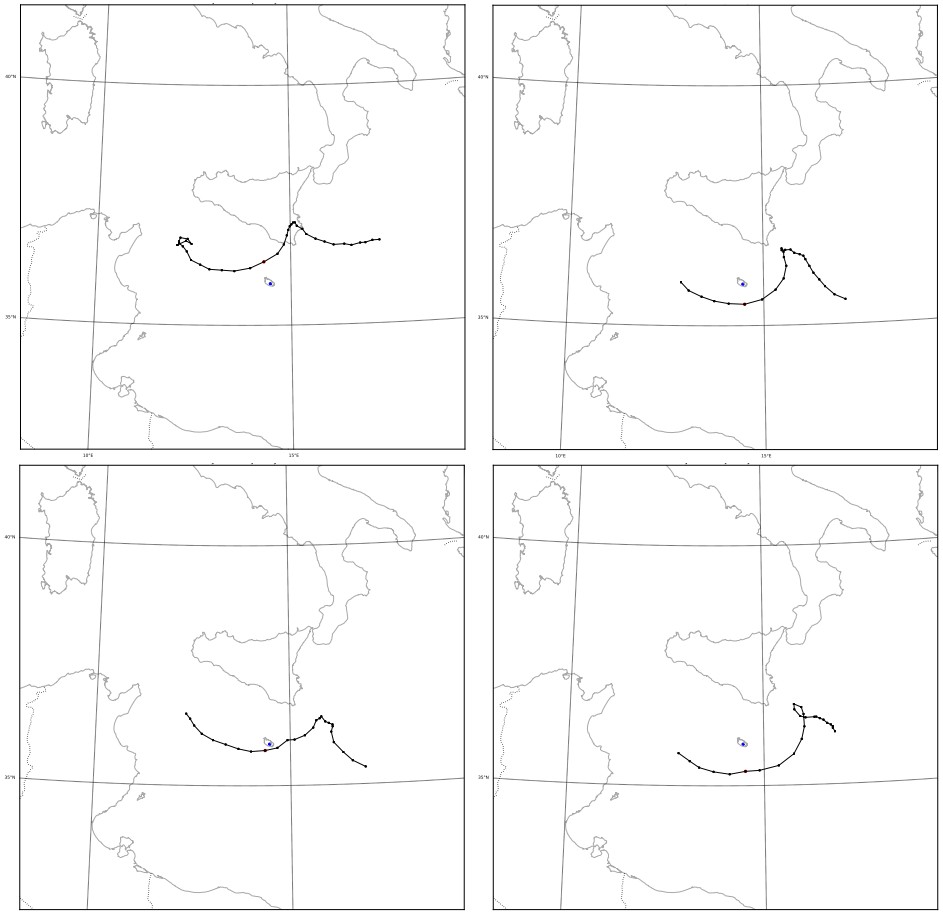

**Figure 12.** Best cyclone's trajectory depicted by each experiment (a) NODA, (b) SYN, (c) RSAMV and (d) CNTRL from 11 UTC 07 November to 12 UTC 08 November 2014.

To assess the error associated to the cyclone's trajectory, the distance between the ensemble mean center of the simulated cyclone and the observed one are computed for each time step (Fig. 13). Before the moment of maximum intensity (i.e.,

approximately at 18 UTC 7 November) NODA experiment depicts a mean track error lower than the data assimilation experiments. However, as we get closer to the maximum intensity of the cyclone, data assimilation experiments depict lower error values than NODA until 00 UTC 8 November. In this period of time, mean track error values from the data assimilation experiments become indistinguishable between them. After 00 UTC 8 November, the errors associated with the data assimilation experiments start to grow and the error associated with NODA start to decrease until the end of the simulation. Regarding the ensemble spread of each experiment (shaded areas in Fig. 13), it is showed that all data assimilation experiments have larger spread than the NODA.

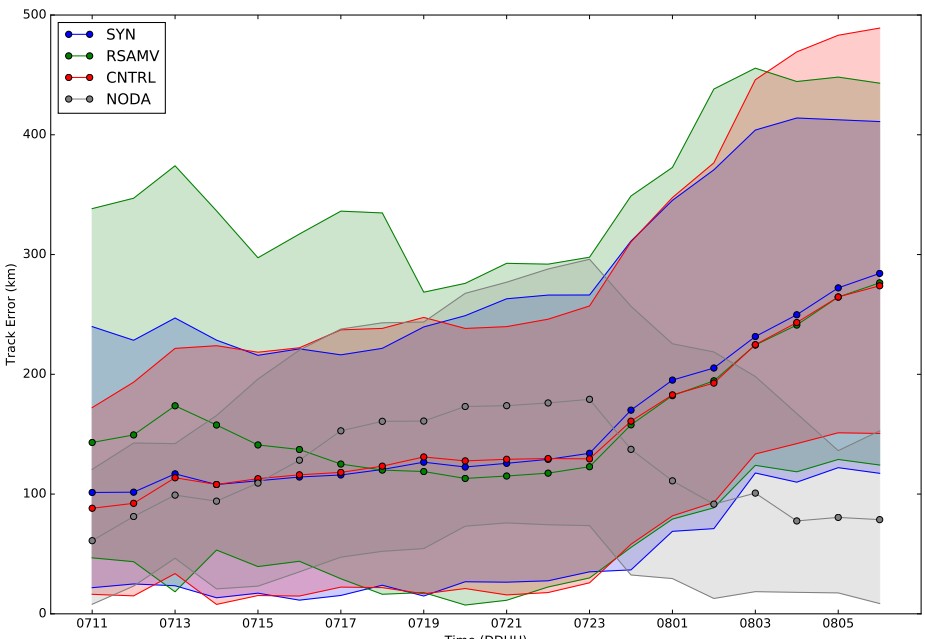

**Figure 13.** Ensemble mean (solid lines) and ensemble spread (shaded areas) of track error (km) associated to NODA_0700, SYN, RSAMV and CNTRL experiments from 11 UTC 07 November to 12 UTC 08 November 2014.

An alternative way of quantitatively assessing the goodness of the track forecasts is also obtained by computing the probability density function $\rho$ of the center of the cyclone at each time step for each ensemble member using a *Kernel Density Estimation* (KDE) statistic method (Scott, 2015; Silverman, 2018). Once the probabilities were computed for each time step, we calculated the integrated probability density using the *additive law of probability* (see Carrió and Homar (2016) for further details). In this way, we obtained the probability distribution of the cyclone center occurrence from 11 UTC 07 November





to 12 UTC 08 November. Finally, to assess how accurate are the different experiments depicting the medicane's trajectory in comparison with the trajectory observed, we computed a line integral of this probability $\rho$ over the observed trajectory using:

$$PCC = \int_{OP} \rho \, dl \qquad (1)$$

where PCC is the accumulated forecasted probability along the actual cyclone track and OP refers to the observed medicane's path. Results indicate that the data assimilation experiments render higher values of probability of having cyclone center

occurrence close to the observations than the NODA experiment (Fig. 14). In particular, SYN, RSAMV and CNTRL experiments show probabilities of 0.6018, 0.8078 and 0.5592, respectively, in comparison with a probability of 0.4002 obtained by NODA. Note that this result is in agreement with the track error results showed in Fig. 13.

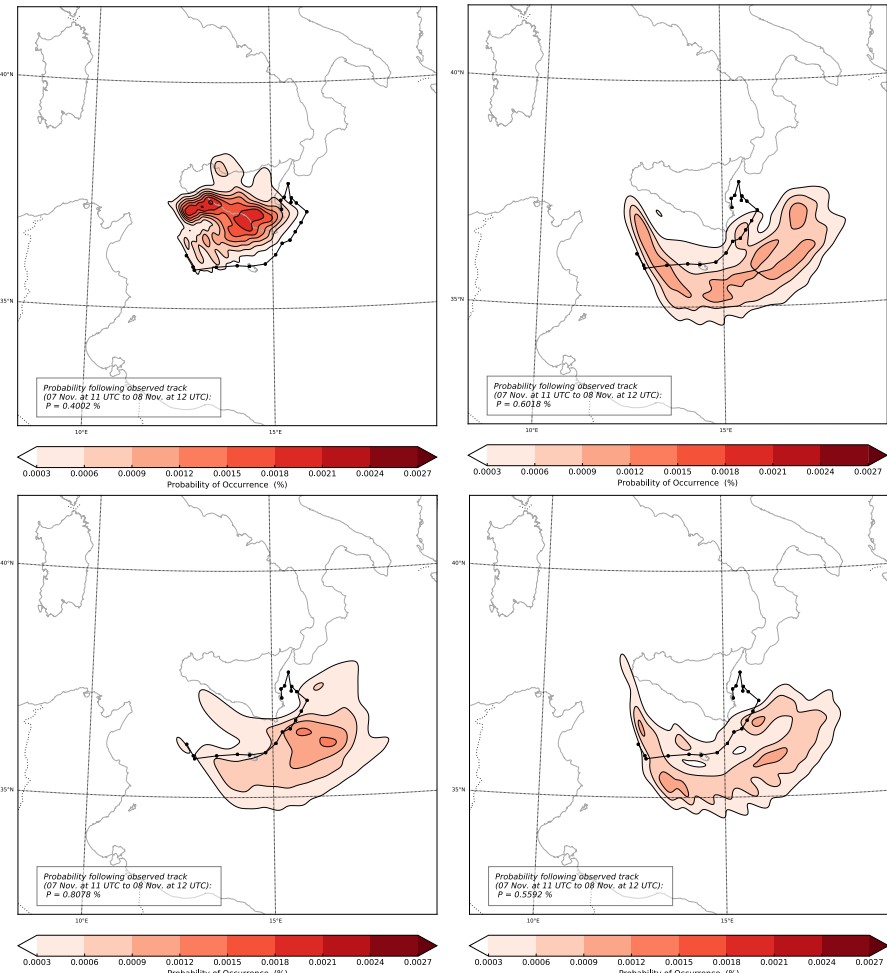

**Figure 14.** Probability of cyclone center occurrence computed using Gaussian KDE technique for (a) NODA, (b) SYN, (c) RSAMV and (d) CNTRL experiments from 11 UTC 07 November to 12 UTC 08 November 2014.





### 4.2.2 DA impact on Qendresa's Intensification forecast

Taking into account the inherent difficulty of the models in properly predicting the intensity of TCs, the effect of assimilating
conventional and RSAMV observations is explored. In this case, the lack of *in-situ* observations over the region where Qen-
dresa took place is the main challenge to properly verify the cyclone's intensity forecasts in a Lagrangian sense (following
the medicane evolution). Instead, we took advantage of the fact that the medicane crossed over Malta island, where METAR
instruments registered a pressure drop greater than 20 hPa in 6h, reaching a minimum surface pressure value of 985 hPa. In this
context, to assess the skill of the different numerical experiments, the METAR information from the Malta airport was used.
Specifically, the surface pressure evolution measured by the METARs was compared against the obtained from the ensemble
members of each experiment. To achieve this comparison the surface pressure evolution of the closest trajectory point to Malta
airport for each ensemble member was used and then compared with the observations.

Results from NODA experiment show that the ensemble mean fits the observations accurately during the first hours of the
forecast, from 00 UTC to 15 UTC 7 November (Fig. 15). However, during the intensification phase, the ensemble mean identi-
fies the surface pressure minimum at the same time it was observed but it does not reach the 985 hPa minimum observed value.
Then, the cyclone starts its dissipation phase in which the ensemble mean is not well adjusted to the observations between
19 UTC 7 November to 06 UTC 8 November. A closer inspection to the ensemble members shows that the surface pressure
minimum is reached at different times between 12 UTC 7 November to 05 UTC 8 November. In other words, the minimum
surface pressure depicted by the NODA ensemble system has a large temporal spread, with most of the ensemble members
simulating a minimum pressure earlier than the observed, but others showing some delay. It is also important to note that some
of the ensemble members reach surface minimum pressure values similar to the observations. In contrast, SYN experiment
reduces significantly the above-mentioned temporal spread showing most of the ensemble members laying near the observed
barogram (Fig. 15b). In this case, the assimilation of *in-situ* conventional observations has contributed in a better prediction of
the minimum surface pressure.

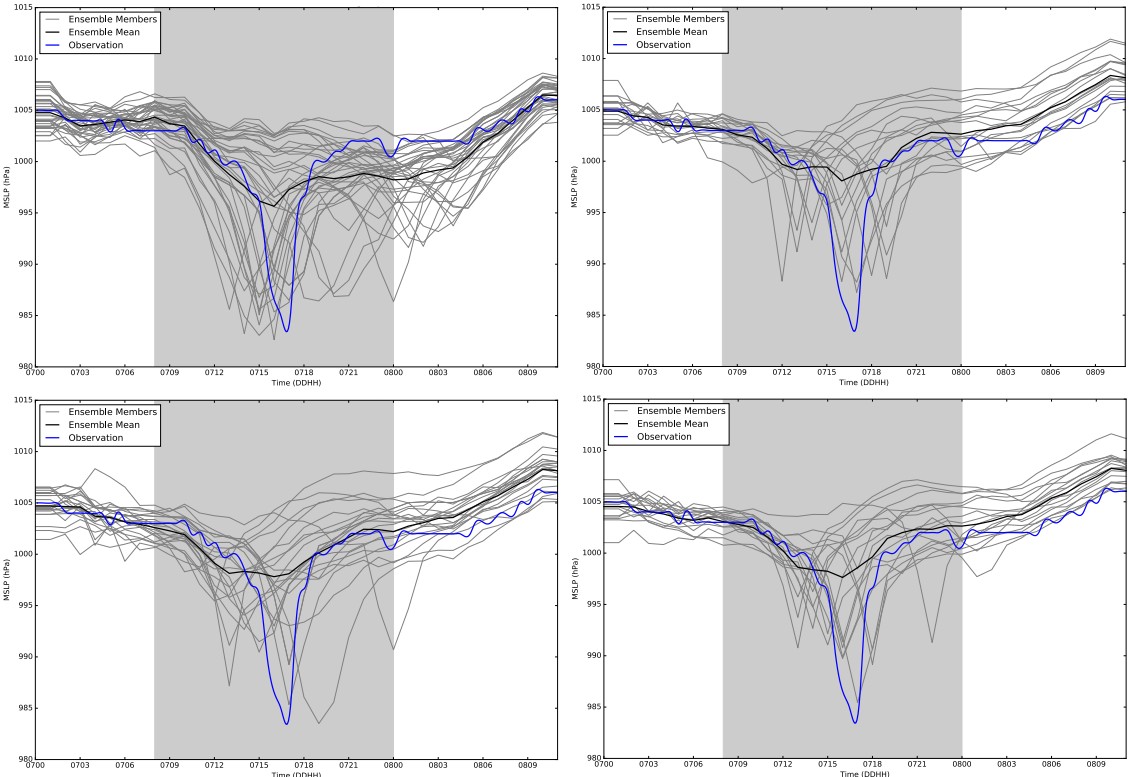

**Figure 15.** Ensemble (grey solid lines) and ensemble mean (black solid lines) surface pressure evolution at the closest grid point to Malta for (a) NODA, (b) SYN, (c) RSAMV and (d) CNTRL experiments. Surface pressure registered by METARs in Malta's airport is also depicted (blue solid lines). Grey shaded area represents the most intense period of the medicane which we are most interested in.

A closer inspection at the time the minimum surface pressure was observed (i.e., 17 UTC 7 November), reveals that several ensemble members depict the minimum pressure at the same time, although the simulated cyclone is shallower than the observed. When only RSAMV observations are assimilated, some members simulate deep cyclones, but again with great vari-

ability among ensemble members (Fig. 15c). It seems that the assimilation of only RSAMV observations is not enough to improve the relevant low-level atmospheric structures. Finally, CNTRL experiment shows similar results to the SYN simulation, but in this case the ensemble members that correctly simulate the maximum depth of the cyclone at 17 UTC show lower surface pressure values than the ones observed by the SYN experiment (Fig. 15d). The ensemble spread of CNTRL is also reduced in comparison with the rest of numerical experiments.


To quantitatively assess the goodness of these results we decided to use the *lagged correlation* between the ensemble members and the observations (Fig. 15). The correlation measures how the V-shape of the surface pressure evolution signal fits the observations from METARs, accounting for its temporal shifting. In this sense, a correlation of 1 would mean that the specific ensemble member has the same V-shape pressure evolution than the observations and also that the minimum for both of them




is found at the same time. Taking into account that some ensemble members deepen faster or slower than observed, we compute *lagged correlations* between ensemble members and observations (Fig. 16). For NODA experiment, the correlation of the ensemble members and the observations is maximum when a delay of 1h is applied to the forecasts (Fig. 16a). In this case, the correlation value for the median of the ensemble is approximately 0.4, with some ensemble members depicting a correlation value larger than 0.9 and others a smaller value than -0.2. For the SYN experiment, the maximum correlation values for the

ensemble members occur when a temporal delay of 1h is applied (Fig. 16b). In this case, the ensemble median is approximately 0.4 and the third quartile is now higher than in the NODA experiment. When RSAMV are the only observations assimilated, it can be seen that the highest correlation is found for a temporal lag of -1h (Fig. 16c). However, in this case, the median of the ensemble shows a correlation value closer to 0.5. Finally, the CNTRL experiment shows again a maximum correlation for a temporal lag of -1h, with a correlation value of 0.5 for the median of the ensemble and maximum correlation values of close

to 1 for some of the ensemble members (Fig. 16d). All the above results indicate that CNTRL is the experiment which best verifies against the observations.

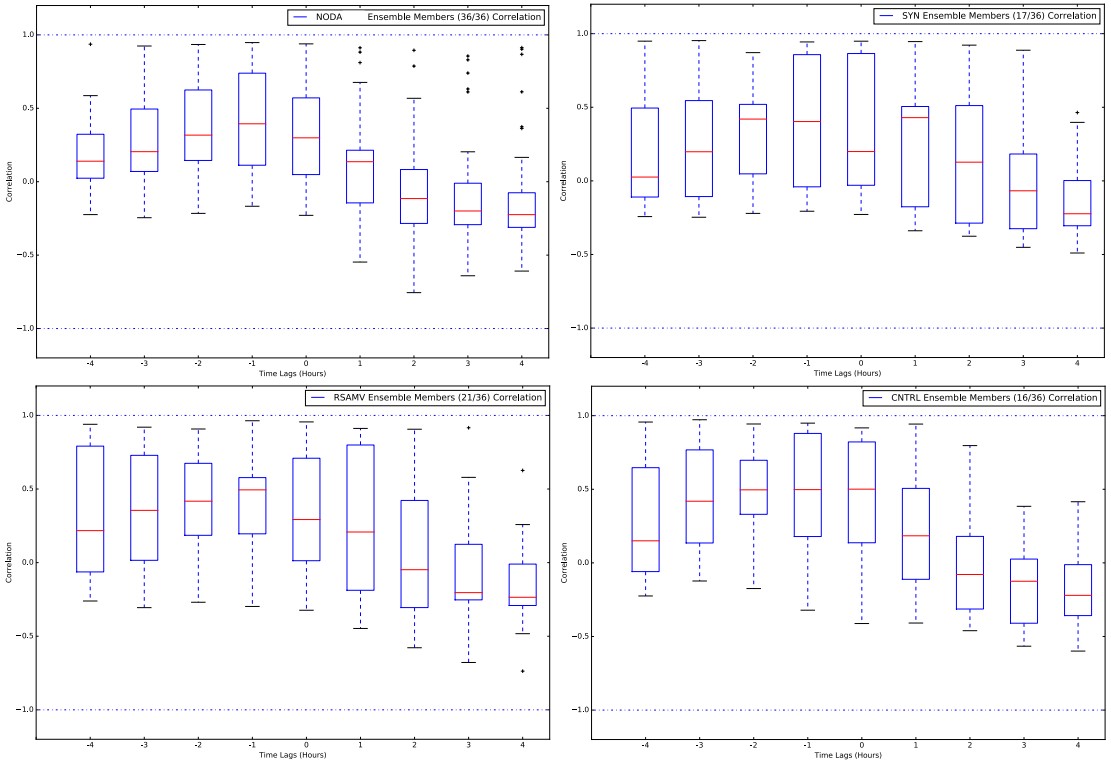

**Figure 16.** Whisker plots depicting the lagged correlation values between the observations and the ensemble members for the different experiments (a) NODA, (b) SYN, (c) RSAMV and (d) CNTRL. The correlation is calculated taking into account that the V-shape signature corresponding to the observations is shifted 4 hours to the left and 4 hours to the right.


## 4.3    A deeper understanding of the effects of EnKF on Qendresa's forecast

In order to better understand how the assimilation of different types of observations using the EnKF affects the forecast results of this low-predictable medicane event, we analyze different meteorological fields among the numerical experiments performed in this study and we compare them against ERA5 reanalysis, which accurately depicts Qendresa's track and intensity. The good agreement of ERA5 and the observations is because ERA5 provides high-temporal resolution fields (i.e., hourly) obtained from the assimilation of vast amounts of observations (most of them satellite radiances), although they are found on a 30 km grid resolution mesh. For the sake of brevity, here we showed these comparisons at two specific times, 00 UTC and 18 UTC on 7 November 2014. We chose 00 UTC because is the initial condition state of all forecasts. Then, we focused on 18 UTC because at that time, the cyclone reached its maximum intensity over Malta island.

First, the MSLP obtained from the different numerical experiments were analyzed in comparison to ERA5 analysis (Fig. 17). Among all the experiments, NODA was the only that depicts an incipient low pressure system resembling the one in ERA5. Regarding the data assimilation experiments, shallower and broader low pressure systems which are also shifted northwards towards the western part of Sicily, were identified. Differences between ERA5 and the data assimilation experiments clearly show a circular negative region of MSLP offshore Tunisia, indicating that these experiments clearly fail in performing an accurate representation of the initial conditions of the atmosphere for this event. At 18 UTC, the only experiment performing a deep cyclone structure is again NODA. However, a MSLP dipole over Malta from the differences between NODA and ERA5 indicates that the cyclone simulated by NODA is located north-westwards from the ERA5 cyclone. Instead, data assimilation experiments predict less intense low pressure systems that are located southeaster of Malta. Thus, initial conditions obtained from the assimilation of *in-situ* conventional and RSAMV observations are facilitating the increase of the cyclone's phase velocity, but in contrast, initial conditions from NODA are reducing such velocity.



**Figure 17.** Comparison of the MSLP between the numerical experiments and the ERA5 analysis at 00 UTC (initial conditions) and 18 UTC (maximum intensity registered at Malta). NODA, SYN, RSAMV and CNTRL experiments are shown in the first, second, third and fourth row, respectively. MSLP simulated by the different experiments at 00 UTC and 18 UTC are shown in the first and third column, respectively, and the differences between ERA5 and the experiments are shown in the second and fourth columns, respectively.

Taking into account the fact that it was showed before that the predictability of this medicane event is strongly associated with the upper-level dynamics, it was also analyzed the geopotential height at 500 hPa and the potential vorticity at 300 hPa.

475

Regarding the geopotential height at 500 hPa, ERA5 initially shows a strong negative tilted trough with two geopotential minimum centers, one over the eastern part of the Balearic Islands and the other located over the south of Tunisia (Fig. 18). All numerical experiments correctly represent the negative tilted trough, but only NODA performs the above-mentioned two


minimum centers of geopotential height with high accuracy. The data assimilation experiments represent a less intense trough

480 (blue shaded areas), specially over the minimum located over Tunisia.

(a)  (b)  (c)  (d)

(e)  (f)  (g)  (h)

(i)  (j)  (k)  (l)

(m)  (n)  (o)  (p)

-120  -90  -60  -30  0  30  60  90  120
Geopotential (mgp)

**Figure 18.** Comparison of the geopotential at 500 hPa between the numerical experiments and the ERA5 analysis at 00 UTC (initial conditions) and 18 UTC (maximum intensity registered at Malta). NODA, SYN, RSAMV and CNTRL experiments are shown in the first, second, third and fourth row, respectively. 500 hPa-geopotential is performed by the different experiments at 00 UTC and 18 UTC are shown in the first and third column, respectively, and the differences between ERA5 and the experiments are performed in the second and fourth column, respectively.

At 18 UTC, a cutoff is predicted by all experiments, being deeper in the NODA than any data assimilation experiments. However, significant differences in the location of the cutoff are found between these experiments. In the NODA experiment, the cutoff is centered approximately over the central part of Sicily, whereas ERA5 locates the center of the cutoff between the




eastern part of Sicily and Malta. The other experiments share the same characteristics: the simulated cutoff is faster (eastwards) than the analysed in ERA5. This can be easily seen looking the geopotential dipoles in Fig. 18(d,h,l,p), which is located southern Sicily.

Finally, due to its primary role in the cyclogenesis dynamics, we also analysed the PV at 300 hPa for the entire set of numerical experiments (Fig. 19). In general, the PV anomaly predicted by the different experiments is very similar to the PV shown in ERA5, but with smaller amplitude and weaker PV gradients. The main difference between the data assimilation experiments and NODA is that, the PV streamer in the latter is more intense. It is important to note that there is no significant differences among the PV structures simulated by the different data assimilation experiments. In the case of SYN, this is likely attributable to the lack of impact at such altitudes from the assimilation of a reduced set of conventional observations. In the case of the RSAMV, the effect is also very small. This is mainly attributed to the fact that a moderate horizontal covariance localization length scale of 50 km, which is necessary to avoid having correlated observation errors, is used. This, together with the fact that only a few RSAMV observations were reported at 300 hPa, are the main reasons for this experiment to not to change significantly the PV trough. At 18 UTC, the analysed PV streamer shows a very intense circular PV structure ($\sim 10$ PVU) centered over Malta island, whereas the numerical experiments predict a wide area of PV with maximum values of 7 PVU over southern Greece.



**Figure 19.** Comparison of PV at 300 hPa between the numerical experiments and the ERA5 analysis at 00 UTC (initial conditions) and 18 UTC (maximum intensity registered at Malta). NODA, SYN, RSAMV and CNTRL experiments are shown in the first, second, third and fourth row, respectively. PV at 300 hPa is performed by the different experiments at 00 UTC and 18 UTC are shown in the first and third column, respectively, and the differences between ERA5 and the experiments are performed in the second and fourth column, respectively.

## 5 Conclusions

The present study aimed to explore the influence of assimilating RSAMV in combination with *in-situ* conventional observations to enhance the forecast of a rare-type of Mediterranean tropical-like cyclone, known as medicanes, which are also known to produce high socio-economic impacts to populated coastal regions in the Mediterranean basin. The medicane discussed here took place over the sea between 6-8 November 2014 and severely affected Malta island. Specifically, this weather event was





selected due to is low-predictable behavior (i.e., poorly forecasted by current numerical weather prediction models) probably
because their initial conditions are poorly estimated due to the lack of observations over the sea. Although data assimilation
techniques have been applied to predict tropical cyclones or typhoons, they have not been applied (as the author is aware) to
medicanes, which are much less predictable events. The analysis obtained from the EnKF (new initial conditions) are expected
to provide a more accurate representation of the mesoscale environment, and thus forecasts started from this initial condition

are also expected to provide more accurate predictions. It is noteworthy that real case data assimilation studies typically as-
similate observations from the moment that the phenomena of interest is well observed and developed. In the case of tropical
cyclone data assimilation studies, observations are assimilated when the tropical cyclone is already formed. However, in this
study we are interested in improving, from an operational point of view, the pre-convective environment before this medicane
event initiated and improve in this sense the warning systems associated with this phenomena.


To study the effect of the assimilation of conventional and RSAMV observations a set of four numerical experiments that
provide initial conditions at 00 UTC on 7 November was designed. In the first experiment, SYN, only *in-situ* conventional
observations were assimilated. In the second experiment, RSAMV, only *Rapid Scan Atmospheric Motion Vectors* observations
were assimilated. In the third experiment, CNTRL, both conventional and RSAMV observations were assimilated and finally,

in the NODA, no data assimilation was performed. Results of these experiments were classified in two groups referring to the
cyclone's track and intensity. Regarding the track of the cyclone, results indicated that in general, ensemble members did not
predict accurate trajectories, being the NODA, the experiment that worse simulated the cyclone track and being the RSAMV,
the one that better predicts the observed trajectory during the mature phase of the medicane. However, cyclone's trajectories
produced by the different data assimilation experiments were shifted towards the east, without performing the track loop ob-

served through satellite imagery. This fact could be associated with the upper level dynamics that acts as a triggering mechanism
of convection, as the comparison of the different experiments with ERA5 shows. In terms of the intensity of the cyclone, the
CNTRL experiment was found to be the best simulation predicting the intensity of the medicane using lagged correlation verifi-
cation, with a correlation median value of nearly 0.5 and some of the ensemble members depicting correlation values close to 1.

Analyzing the ensemble mean evolution of the surface pressure for all the experiments and comparing them against ERA5
analysis we found that data assimilation experiments (SYN, RSAMV and CNTRL) were simulating the center of the cyclone
eastwards the center of the ERA5 cyclone. To better understand this behavior, we analyzed the upper level dynamics through
the geopotential at 500 hPa and the PV at 300 hPa. Geopotential at 500 hPa simulated by the data assimilation experiments
show a well-defined cutoff displaced also towards the east of the cutoff depicted by ERA5. Then, PV at 300 hPa depicted a

broad and weak structure centered between south Italy and south Greece, in contrast to the clear intense structure centered
over Malta depicted by ERA5. In other words, the assimilation of RSAMV did not modify significantly such upper-level
structures which are crucial for the correct prediction of this medicane event. We hypothesize that the fact that this result could
be attributable to the fact that the availability of RSAMV observations in the upper-levels were marginal and thus, the data
assimilation system was not able to properly modify the upper-level dynamics.



*Author contributions.* All the work presented in this study (e.g., design and performance of the numerical simulations, interpretation of results and writing of the manuscript) has been carried out by D. S. Carrió.

*Competing interests.* The authors declare that they have no conflict of interest.

*Acknowledgements.* This research is framed within the ARC Centre of Excellence for Climate Extremes (CE170100023). The author thankfully acknowledge Meteo-France for supplying the data and HyMeX database teams (ESPRI/IPSL and SEDOO/OMP) for their help in ac-
cessing the data. The author also acknowledge the computer resources at MareNostrum IV and the technical support provided by Barcelona Supercomputing Center (RES-AECT-2017-1-0014, RES-AECT-2017-2-0014) that allowed to perform the high-resolution simulations presented in this study. Thanks to Dr. Louis Wicker and Kent Knopfmeier from the National Severe Storm Laboratory for his technical assistance implementing the EnKF system. Special thanks to Prof. Víctor Homar for his valuable comments to improve the quality of this study.



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
