# Peer review of "Improving the predictability of the Qendresa medicane by the assimilation of Conventional and Atmospheric Motion Vector observations. Storm-scale analysis and short-range forecast."

_Natural Hazards and Earth System Sciences, 2022_

## Author Comment (AC1)

**The University of Melbourne**
Diego Saúl Carrió Carrió
School of Geography, Earth and Atmospheric Sciences and
Centre of Excellence for Climate Extremes
Parkville, 3010
Victoria, AUSTRALIA
diego.carriocarrio@unimelb.edu.au
Melbourne, April 26, 2022

Dr. Joanna Staneva
Editor– Natural Hazards and Earth System Sciences

Dear Joanna Staneva,

Please find attached the revised version of the manuscript **NHESS-2022-58** entitled *"Challenges assessing the effect of AMVs to improve the predictability of a medicane weather event using the EnKF. Storm-scale analysis and short-range forecast"*.

I have carefully examined the constructive suggestions made by the reviewers and I have taken full account of their comments. Therefore, the main results of the work are now better described and emphasized. The following is a point-by-point response to the comments and inquiries made by the first reviewer. I thank the reviewer for their comments and believe the manuscript is now greatly improved both in terms of clarity and readability. We believe that the new version of the manuscript, which have been improved significantly, will help the reader to better understand the work presented here.

With best regards,

Diego Carrió Carrió

**ANSWERS TO THE REVIEWER**

**Reviewer #1:**

*The manuscript presents a study to evaluate the impact of assimilation of atmospheric motion vectors and conventional observations with an ensemble Kalman filter to improve forecast of medicane Qendresa.*

*The manuscript is well written and the numerical experiments well designed.*

**Major comments:**

*Main remarks are as follow:*

1. *The introduction requires a brief general presentation of numerical weather prediction and data assimilation, before any detailed discussion.*

I thank the reviewer for pointing out this. I totally agree with the reviewer that a brief general description of numerical weather prediction along with data assimilation will help the reader better contextualize the work presented here.

To this purpose I have added the following sentences at the beginning of the *Introduction* section:

"*The Western Mediterranean region is often affected by high impact weather phenomena (e..g, tornadoes, hail storms, tropical cyclones, or flash floods, among others) which produce huge economic losses and fatalities. Although numerical weather prediction models have significantly improved during the last years, the proper prediction of such extreme weather events (timing and location) in the Western Mediterranean remains a challenge. Many extreme weather events affecting the Mediterranean coastlands initiate over the sea, where in-situ observations are scarce. This lack of information impacts negatively on the representation of the initial state of the atmosphere, and consequently on the accuracy of the numerical forecasts (Wu et al., 2013). Over the last years, different sophisticated methods have been designed and implemented to improve the estimation of the atmospheric state in numerical weather prediction models from both in-situ and remote sensing instruments, such as Doppler radars or meteorological instruments on-board satellites (Rabier, 2005; Palmer and Hagedorn (2006); Shen et al., 2016; Geer et al., 2018). These methods are known as Data Assimilation techniques (e.g., Daley, 1991; Evensen, 2009b; Kalnay, 2002), which basically combines information from numerical weather prediction forecasts with all the available observations to create a new set of initial conditions that better represent the current state of the atmosphere.*"

References:

*Xinrong Wu, Shaoqing Zhang, Zhengyu Liu, Anthony Rosati, and ThomasL. Delworth. A study of impact of the geographic dependence of observing system on parameter estimation with an intermediate coupled model. Climate Dyn., 40(7-8), 2013.*

*Florence Rabier. Overview of global data assimilation developments in numerical weather-prediction centres. Quart. J. Roy. Meteor. Soc., 131(613), 2005.*

*Tim Palmer and Renate Hagedorn. Predictability of weather and climate, chapter Observations, assimilation and the improvement of global weather prediction-some results from operational forecasting and ERA-40. Cambridge University Press, 2006.*

*Feifei Shen, Jinzhong Min, and Dongmei Xu. Assimilation of radar radial velocity data with the wrf hybrid etkf–3dvar system for the prediction of hurricane ike (2008). Atmospheric Research, 169, 2016.*

*Eugenia Kalnay. Atmospheric modeling, data assimilation and predictability. Cambridge university press, 2002.*

*Evensen, G. Data assimilation: The ensemble Kalman filter(2nd ed.). Berlin and Heildelberg, Germany: Springer-Verlag, 2009b.*

*Roger Daley. Atmospheric data analysis, cambridge atmospheric and space science series. Cambridge University Press, 6966(25):809–822, 1993.*

*Geer, A. J., Lonitz, K., Weston, P., Kazumori, M., Okamoto, K., Zhu, Y., ... & Schraff, C. (2018). All-sky satellite data assimilation at operational weather forecasting centres. Quarterly Journal of the Royal Meteorological Society, 144(713), 1191-1217.*

2. *When referring to present weather prediction systems/models more details should be included: are these operational forecasts or case studies? Are these global coarse resolution or mesoscale convection permitting models? Do they assimilate observations and how?*

I really appreciate the reviewer's comment about this point. The initial idea was to highlight that even using the most recent numerical weather model versions at high grid resolution we still have problems to accurately predict the intensification and trajectory of this small-scale Mediterranean cyclones, which are mainly initiated over sparse observational regions. To avoid misinterpretation of this sentence to the reader we have rewritten the sentences that appears "current numerical weather models…" as:

Line L6: "…*the chaotic behavior inherent to current numerical weather prediction models*" => "…*the chaotic behavior inherent to **numerical weather prediction models***"

L94-95: "*More precisely, the correct prediction of both the northward loop trajectory followed by Qendresa and its intensification still remain a major challenge for most current numerical weather models.*" => "*More precisely, the correct prediction of both the northward loop trajectory followed by Qendresa and its intensification still remain a major challenge for most **current mesoscale convection-permitting numerical weather models***."

L130-131: "*It is hypothesized that the low predictability of Qendresa is likely associated to the difficulty of current numerical weather models to depict…*" => "*It is hypothesized that the low predictability of Qendresa is likely associated to the difficulty of **current mesoscale convection-permitting numerical weather models** to depict…*"

L139-140: "*This loop-like trajectory is the main feature that current numerical models do not accurately predict mainly due to a miss representation of the upper-level dynamics*" => "*This loop-like trajectory is the main feature that **current convection-permitting numerical models** do not accurately predict mainly due to a miss representation of the upper-level dynamics*"

L504-505: "*Specifically, this weather event was selected due to is low-predictable behavior (i.e., poorly forecasted by current numerical weather prediction models) probably because their initial conditions are poorly estimated due to the lack of in-situ observations over the sea.*" => "*Specifically, this weather event was selected due to is low-predictable behavior associated to the poorly initial conditions estimation of the atmosphere over the sea.*"

3. *The ensemble was designed to include different sets physical parameterizations among the different ensemble members (if I understood correctly). However, no discussion on how different sets of parameterizations performed is discussed. Can you identify sets of parameterizations that lead to better/worse results in a systematic way? If so, I suggest also drawing them with a different color in the plots.*

The ensemble of forecasts used in this study was generated using initial and boundary conditions from the global EPS-ECMWF and then to add more diversity to the ensemble different combinations of parametrizations were also used for each of the ensemble members. This approach is widely used in the National Severe Storms Laboratory (NSSL-NOAA, Oklahoma,USA) and NCAR (USA), where the author learnt this method among others. At the end of the first paragraph of section 3.1 it is also stated that more details about how the ensemble was generated is explained in the paper by Carrió and Homar, 2016. Table 1, shows the different combination of parametrizations used for each ensemble member.

In this study I focus on the performance of the EnKF to improve the predictability of this extreme and low predictable weather event. To this purpose, the ensemble design plays an important role so we want to avoid having an under dispersive ensemble, which will hamper the data assimilation performance. After testing different sets of parametrizations I ended with an ensemble configuration that presents the statistical properties I was interested in. The ensemble set up is the one described in the manuscript.

In this sense, investigating the performance of individual ensemble members according with its parametrizations is beyond the scope of this study. Adding this information will no contribute to understand better results of this study and could distract the attention of the reader of other crucial aspects of this study. The main aim of this study is to investigate the impact of the assimilation of AMVs to improve pre-convective environment of this high-impact weather event. We do not focus on seeking the best numerical simulation of this event.

Nevertheless, I think that following the reviewer's comment I could improve the section adding the following sentences at the end of section 3.1 (*Numerical Weather Model*):

"*Note that the main objective of this paper is not to investigate the sensitivity of grid resolution and the role of different parameterizations to the Qendresa forecast skill.*

*However, if one is interested in the potential effect of these aspects to Qendresa forecast, please see recent work from Pytharoulis et al., 2018 and Mylonas et al., 2019."*

4. *The use of the global model for initial and boundary conditions is mentioned in the manuscript, however, no discussion is provided regarding the improvement that I supposed was achieved by the WRF model and WRF-DART simulations, both due to better resolution and assimilation at high resolution. This should be discussed in the manuscript and lines depicting the global model results should be added in the plots.*

The comparison and discussing that the reviewer is making about the global model is fair and indeed, that discussion is already present in the paper. To investigate the potential of assimilating the different kind of observations I was interested in (i.e., conventional and AMVs) I designed 4 experiments: (a) NODA, in which no observations are assimilated, (b) SYN, only conventional observations are assimilated, (c) RSAMVs, only AMVs observations are assimilated and (d) CNTRL, both conventional and AMVs are assimilated. These experiments share the same spatial grid resolution, so I can avoid attributing benefits of the results to such aspect and I can focus more on the observations assimilated.

It is also important that it would not be fair to compare the DA simulations against the global model, which is the EPS-ECMWF that runs at ~31 km for the period when Qendresa took place. Our NODA experiment is the equivalent to global model downscaled at the resolution of the rest of experiments (i.e., 4 km). That was the main reason behind the design of the NODA experiment, to be able to compare its performance with the DA experiments and be able to assess the impact of assimilating different observations, all of them using the same grid spacing resolution.

**Minor remarks:**

*Line 8: with operational models? At what resolution? Lower than your resolution?*

As far as the author is aware, accurate prediction in terms of intensity and trajectory of Qendresa, especially the observed loop in the trajectory eastern of Sicily, was very challenging. Operational models that tried to predict such event fail and other researchers in the community (some of them exposed at conferences but never published) that run their own models, such as WRF, also failed. In terms of spatial grid resolution, there were simulations with higher and lower grid resolution compared to the one we used in this study, and none of them succeeded.

This issue was solved in the above answer to point number 2.

*Line 16: Is your high resolution the main factor of improvement or the data assimilation?*

Results of this study, after comparing the DA experiments (i.e., SYN, RSAMV and CNTRL) against the simulation where no data were assimilated (i.e., NODA), all of them using the same grid resolution, reveal that the main factor of improvement comes from the data assimilation procedure.

*Line 18: A short introduction of NWP and data assimilation should be provided before directly jumping into detailed review of data assimilation literature*

I agree with the reviewer and the revised version of the manuscript contains the following lines at the beginning to the introduction section:

*"The Western Mediterranean region is often affected by high impact weather phenomena (e..g, tornadoes, hail storms, tropical cyclones, or flash floods, among others) which produce huge economic losses and fatalities. Although numerical weather prediction models have significantly improved during the last years, the proper prediction of such extreme weather events (timing and location) in the Western Mediterranean still remains a challenge. Many extreme weather events affecting the Mediterranean coastlands initiate over the sea, where in-situ observations are scarce. This lack of information impacts negatively on the representation of the initial state of the atmosphere, and consequently on the accuracy of the numerical forecasts (Wu et al., 2013). Over the last years, different sophisticated methods have been designed and implemented to improve the estimation of the atmospheric state in numerical weather prediction models from both in-situ and remote sensing instruments, such as Doppler radars or meteorological instruments on-board satellites (Rabier, 2005; Palmer and Hagedorn (2006); Shen et al., 2016; Geer et al., 2018). These methods are known as Data Assimilation techniques (e.g., Daley, 1993; Evensen, 2009b; Kalnay, 2002), which basically combines information from numerical weather prediction forecasts with all the available observations to create a new set of initial conditions that better represent the current state of the atmosphere"*

*Line 36: This part of the sentence is not clear. May be: Due to the limited oceanic and maritime coverage of meteorological radars,....*

I thank the reviewer for pointing out this. We have modified the sentence according with the suggestion provided by the reviewer.

*Line 39: May be: observational coverage of*

I Agree. We have applied the reviewer suggestion.

*Line 50: These are not so recent. Are there any more recent references?*

I thank the reviewer's suggestion. Yes, there are more recent references. The revised version of the manuscript contains the following recent references about bias correction.

*Kumar, S. V., Reichle, R. H., Harrison, K. W., Peters-Lidard, C. D., Yatheendradas, S., & Santanello, J. A. (2012). A comparison of methods for a priori bias correction in soil moisture data assimilation. Water Resources Research, 48(3).*

*Otkin, J. A., Potthast, R., & Lawless, A. S. (2018). Nonlinear bias correction for satellite data assimilation using Taylor series polynomials. Monthly Weather Review, 146(1), 263-285.*

*Ma, C., Wang, T., Jiang, Z., Wu, H., Zhao, M., Zhuang, B., ... & Wu, R. (2020). Importance of bias correction in data assimilation of multiple observations over eastern China using WRF-Chem/DART. Journal of Geophysical Research: Atmospheres, 125(1), e2019JD031465.*

*Line 54: It should be stated that one of the weaknesses of retrievals is the need for a first guess or additional atmospheric information, usually coming from short model forecasts that may be in turn too coarse and inaccurate. Some references should be included here.*

I thank the reviewer for pointing this valuable information out. I have added the following sentence in the revised manuscript with some new references:

*L53: "Retrievals are easier to assimilate and interpret because they provide information that can be directly related with atmospheric variables, and its assimilation avoids the use of relatively complex RTM. However, one always should consider that atmospheric retrievals are sensitive to many factors, including the first guess that usually comes from a coarse global numerical weather model that could result too inaccurate (Hannon et al., 1996; Li et al., 2000; Zhang et al., 2014)."*

References included:

*Hannon, S. E., L. L. Strow, and W. W. McMillan, 1996: Atmospheric infrared fast transmittance models: A comparison of two approaches. Proceedings of SPIE, 2830, 94–105.*

*Li, J., W. Wolf, W. P. Menzel, W. J. Zhang, H. L. Huang, and T. H. Achtor, 2000: Global soundings of the atmosphere from ATOVS measurements: The algorithm and validation. J. Appl. Meteor., 39, 1248–1268.*

*Zhang, J., Li, Z., Li, J. et al. Ensemble retrieval of atmospheric temperature profiles from AIRS. Adv. Atmos. Sci. **31,** 559–569 (2014). https://doi.org/10.1007/s00376-013-3094-z*

*Line 61: any more recent reference?*

Walker, E., Mitchell, D., & Seviour, W. (2020). The numerous approaches to tracking extratropical cyclones and the challenges they present. *Weather*, 75(11), 336-341.

Choy, C. W., Lau, D. S., & He, Y. (2020). Super typhoons Hato (1713) and Mangkhut (1822), part II: challenges in forecasting and early warnings. *Weather*.

Dorian, T., Ward, B., & Chen, Y. L. (2018). Tropical Cyclone Amos (2016) Forecasting Challenges: A Model's Perspective. *Tropical Cyclone Research and Review*, 7(3), 172-178.

*Line 99: These were not assimilated in previous studies? Are these assimilated in operational forecasts that issued real-time forecasts of Qendresa? If so, they did not improve forecasts? If so, for what reason? Coarse resolution, not efficient data assimilation technique?*

As far as the author is aware AMVs have been not assimilated at this grid resolution for the Qendresa case or any other extreme weather event in the Mediterranean region. Global models, such the ones form the ECMWF tend to assimilate such observations among others. However, the grid resolution for such models is not comparable with the grid resolution used in this study, in which the model is able to resolve much more complex physical processes providing more realistic atmospheric fields. In this study I want to assess the potential of assimilating AMVs. Are the AMVs able to produce a significant improvement to high resolution simulations?

It is difficult to answer if the assimilation of AMVs improve global models or, even more difficult, to what extent they improve the analysis. Global models assimilate AMVs among millions or other types of observations, so one cannot answer this question without performing additional sensitivity experiments. However, in my opinion, for extreme weather events it is crucial to have a high-resolution model running together with an efficient data assimilation method.

In any case, here I want to emphasize that I am not interested in competing with the global model. The motivation of this study was that running a high-resolution model we were not able to reproduce the intensity and track of Qendresa. Due to the limitation of observations available over the region where the medicane took place one appealing observation alternative was to assimilate information about the wind field. According with *Carrió et al., 2017*, improving the upper-level dynamics could contribute to improving the forecast of Qendresa. The assimilation of AMVs offers us this possibility.

*Line 135: can you please further describe how this was done?*

I simply took infrared imagery, where it is easy to identify the eye of the cyclone, and thus, its center. One can "approximately" identify the center of the cyclone for each time stamp infrared imagery is available. Thus, the observation track showed in Fig. 2 only shows the trajectory of the cyclone when the center of the cyclone was identifiable trough infrared imagery.

Following the reviewer suggestion, we have modified that sentence as follows:

*"Due to the lack of in-situ observations present over the maritime region where the medicane took place, infrared imagery was used as a proxy to visually identify the center of the cyclone and estimate the track followed by Qendresa (Fig. 2)."*

*Line 140: which models? Operational? which ones? Other studies? Please cite as necessary.*

I have rephrased the sentence as follows (adding some additional references):

*"This loop-like trajectory poses a clear challenge in terms of numerical predictability, even for current most advance mesoscale numerical weather models (see Carrió et al., 2017, Pytharoulis et al., 2018 and Mylonas et al., 2019)."*

*Line 158: why was one-way nesting chosen?*

I initially tested two-way nesting and I did not find any significant difference from one-way nesting apart from the computational time required to run the two-way nesting in comparison with the one-way nesting (at least for this case). For this reason, all the results shown in the manuscript were obtained using the one-way nesting.

*Line 193: I think that this sentence should appear earlier in the manuscript (where the aim of the study is described) and in the abstract*

We thank the reviewer for noticing this. Following this suggestion, we have deleted this sentence from Line193 and added to the abstract and the introduction sections.

*Line 308: I think that only a short paragraph with aim and main results should be here and the rest as supplementary material.*

I agree with the reviewer that this section could be shortened

*Line 351: Why? It has no data assimilation*

I thank the reviewer for pointing out this. I agree that this sentence is not clear and needs further clarification. The reason behind that sentence is because the initial and boundary conditions from NODA are obtained from the analysis EPS-ECMWF at 00 UTC 7 November, which are obtained from the assimilation of all the available observations (including radiances). Thus, one could expect that the initial conditions at that time could be better than the initial conditions obtained from the rest of DA experiments after cycling 12 hours and only assimilating conventional and AMV observations. Here it is showed the importance of assimilating observations using a higher grid spacing.

To avoid misunderstandings, I have rewritten the sentence in the following way:

" F*orecast results from NODA show that most ensemble members do not properly simulate the observed trajectory, particularly the loop-ending on the eastern coast of Sicily, which is produced when the cyclone made landfall and its dissipation phase begin (Fig. 11a)."*

*Line 392: This requires further description. Do the results using this technique add further value? I wonder whether this is really needed. As written now it is unclear to me.*

I agree with the reviewer that this section is not adding new information to the conclusions of these experiments. For this reason, I have deleted the results from the *Kernel Density Estimation*.

*Line 401: All of these are very low probabilities and the kernel method was not explained, therefore the reader cannot understand the importance of these figures*

Following the above comment, I have deleted these results.

*Line 457: Are RSAMV assimilated too?*

As far as the author is aware, only AMVs are assimilated, which are available hourly. RSAMVs are not considered in ERA5 analysis. RSAMVs are the same type of observations but instead of being available hourly, they are available every 20-min. As ERA5 analysis are obtained hourly, there is no need to assimilate them every 20-min.

I have rewritten the sentence to clarify this point raised by the reviewer in the following way:

"*… ERA5 provides high-temporal resolution fields (i.e., hourly) obtained from the assimilation of vast amounts of observations (most of them satellite radiances)…" =>"… ERA5 provides high-temporal resolution fields (i.e., hourly) obtained from the assimilation of vast amounts of observations (most of them satellite radiances), although they are found on a 30 km grid resolution mesh. It is important to note that to obtain ERA5 analysis AMVs have been*

*assimilated. However, the RSAMVs observations assimilated in this study, which are available every 20-min, are not used to produce ERA5 analysis (Hersbach et al., 2020)."*

References:

*Hersbach, H., Bell, B., Berrisford, P., Hirahara, S., Horányi, A., Muñoz-Sabater, J., ... & Thépaut, J. N. (2020). The ERA5 global reanalysis. Quarterly Journal of the Royal Meteorological Society, 146(730), 1999-2049.*

---

## Author Comment (AC2)

**The University of Melbourne**
Diego Saúl Carrió Carrió
School of Geography, Earth and Atmospheric Sciences and
Centre of Excellence for Climate Extremes
Parkville, 3010
Victoria, AUSTRALIA
diego.carriocarrio@unimelb.edu.au
Melbourne, September 23, 2022

Dr. Joanna Staneva
Editor– Natural Hazards and Earth System Sciences

Dear Joanna Staneva,

Please find attached the revised version of the manuscript **NHESS-2022-58** entitled *"Challenges assessing the effect of AMVs to improve the predictability of a medicane weather event using the EnKF. Storm-scale analysis and short-range forecast"*.

I have carefully examined the constructive suggestions made by the reviewers and I have taken full account of their comments. Therefore, the main results of the work are now better described and emphasized. The following is a point-by-point response to the comments and inquiries made by the second reviewer. I thank the reviewer for their comments and believe the manuscript is now greatly improved both in terms of clarity and readability. I believe that the new version of the manuscript, which have been improved significantly, will help the reader to better understand the work presented here.

With best regards,

Diego Carrió Carrió

**ANSWERS TO THE REVIEWER**

**Reviewer #2:**

*The author describes and implements an assimilation system to improve the prediction of medicane Qendresa of 2014 using atmospheric motion vectors (AMVs) and other observations. The author presents the result of four ensemble simulations: one without data assimilation and 3 experiments with data assimilation (SYN, RSAMV, CNTR, section 3.4).*

**Major comments:**

*Main remarks are as follow:*

1. *The author mentions that not all assimilative ensemble members represent the medicane (roughly only the half of the ensemble includes the mediane; exact number of the experiments are on page 20). The manuscript did not mention how many of the ensemble members for the free ensemble model run includes the medicane but judging from figure 11, it seems more ensemble members of the free ensemble represent this medicane. Unfortunately, this is not addressed in the manuscript. It would seem to me a necessary first step, why apparently the assimilation inhibits the generation/developpement of the medicanes and to improve this aspect.*

I really appreciate the reviewer's comment about this point, and I agree that it requires a further explanation. In the case of the NODA experiment, the number of ensemble members that reproduce a cyclone is 29/36, which of course is a bigger number than the rest of data assimilation experiments. The following sentence have been added to the manuscript to provide information about the number of ensemble members of NODA producing the medicane:

*"It is important to note that although most of the ensemble members of NODA (29/36) simulate small vortex circulations, the associated trajectories ("inverted-U" shape) differ significantly from the observations ("U" shape), particularly during the mature stage of the medicane."*

However, note that the tracks depicted by NODA are completely off in comparison with the observations, in particular during the mature stage of the medicane. In addition, they are not able to reproduce the observed trajectory loop in terms of location and time (see comparison in Fig. 12 using the best track selected subjectively). In other words, although NODA reproduces more cyclones, they are not correctly forecasted in terms of trajectory. On the other hand, when all the available observations in this study are assimilated (CNTRL run), the medicane tracks seem to behave more like the observed track, showing the "U" shape characteristic of the observed track. In terms of the error, Figure 13 shows how the track error in the CNTRL experiment is less than NODA during the mature phase, between 16 UTC and 23 UTC on 7 November.

Furthermore, it is also important to note that in this study, NODA is initialized with more recent initial and boundary conditions from the global model valid at 00 UTC 7 November (Fig. 6). However, the data assimilation experiments are initialized using initial and boundary conditions from 12 UTC 6 November, 12 hours before, when the cyclone was not formed. In this sense, one should avoid direct comparison between NODA and DA experiments and be careful with the conclusions that could obtain from this comparison. Here, we are comparing

our data assimilation experiments with NODA, which is initialized with more recent initial conditions from the global model. As it is explained in the manuscript, this comparison is made in this way because we were interested in a comparison from an operational point of view. For example, let's assume that today will take place a severe weather event affecting our local region. What approach should we use: (a) use the most recent initial conditions from global ECMWF (i.e., direct downscaling which does not account for convective-scale observations, such as radars) or (b) should we use the previous analysis from the ECMWF and then assimilate different observations at convective scales, such as reflectivity from radar with high spatial and temporal resolution? This is the motivation behind this comparison, and we have performed other past studies applying the same methodology (see Carrió et al., 2016 or Carrió et al., 2019). For this reason, I believe that we cannot conclude that the assimilation of observations are inhibiting the development of the medicane, because the data assimilation experiments are using different initial and boundary conditions from the global ECMWF, and it is not directly comparable with the NODA results. In fact, NODA initialized at 12 UTC 6 November (which is not shown in the manuscript), which would be a fairer comparison against the data assimilation experiments, shows 18 cyclones, a similar number of cyclones that the ones observed in the data assimilation experiments.

**References:**

*Carrió, D. S., Homar, V., & Wheatley, D. M. (2019). Potential of an EnKF storm-scale data assimilation system over sparse observation regions with complex orography. Atmospheric Research, 216, 186-206.*

*Carrió, D. S., & Homar, V. (2016). Potential of sequential EnKF for the short-range prediction of a maritime severe weather event. Atmospheric Research, 178, 426-444.*

2. *The model validation statistics later are only based on the subset of model members representing the medicane which can result in a misleading interpretation. For the sake of argument, let's take an extreme case when only one ensemble member would represent the medicane track but with good accuracy. The average track would then be simply equal to this medicane trak. In an ensemble would represent 11 members with an mediane, one with a good trak and one with a 10 a biased track, then the average will be worse than the first ensemble average track. In a forecasting scenario you are not sure whether actually a mediane will develop or not, the latter example (ensemble with 11 medicanes) seems to be more informative to me. However, by computing the average only over members including the medicanes favors in this case, the ensemble members with fewer medicanes. To make this more concrete, we can use Figure 15 where the atmospheric pressure at the Malta airport is shown. In a forecasting scenario, your best estimate of the atmospheric pressure would be the ensemble mean using all ensemble members (including those who do not include the medicane) as you do not know yet if the medicane will actually develop or dissipate early. From Figure 15, it even seems that all 36 ensemble members of the free ensemble simulation were used (even those with a very weak depression) which would bias the results even more towards the assimilation simulation where the ensemble members without medicane are excluded.*

I totally agree with the reviewer that only accounting for the ensemble members that reproduce the medicane would lead a misleading interpretation, **if and only if,** the information about the number of ensemble members that it is used to compute this average is not provided. Here, our intention is simply to quantify the error of the cyclone tracks obtained by the different

experiments in comparison with the observations. The problem is that every single experiment is forecasting different number of cyclones, so the comparison in terms of track error is not straightforward. That's why we just use the ensemble members that were forecasting a cyclonic circulation. In terms of the comparison between the data assimilation experiments (i.e., SYN, RSAMV, CNTRL), the number of ensemble members that depicts a cyclone is approximately the same, so comparing their performance is a good approximation. However, the number of ensemble members producing a cyclone in NODA is significantly different, and we should bear in mind this, as the reviewer suggests. For this reason, I have added a sentence in the manuscript to remind the reader that the comparison we are doing here is only fair/valid if we bear in mind all the aspects of the different experiments, such as the number of ensemble members that reproduce a cyclone:

*"To assess the error associated to the cyclone's trajectory, the distance between the ensemble mean center of the simulated cyclone and the observed one are computed for each time step (Fig. 13). Before the moment of maximum intensity (i.e., approximately at 18 UTC 7 November) NODA experiment depicts a mean track error lower than the data assimilation experiments. However, as we get closer to the maximum intensity of the cyclone, data assimilation experiments depict lower error values than NODA until 00 UTC 8 November. During this period, mean track error values from the data assimilation experiments become indistinguishable between them. After 00 UTC 8 November, the errors associated with the data assimilation experiments start to grow and the error associated with NODA start to decrease until the end of the simulation. Regarding the ensemble spread of each experiment (shaded areas in Fig. 13), it is showed that all data assimilation experiments have larger spread than the NODA.* **Because the number of ensemble members between NODA and the DA experiments used to compute the track error differs significantly, special caution should be taken making conclusions comparing NODA and data assimilation results**."

Regarding the reviewer comment from Figure 15 we believe that maybe the idea behind how this figure was made is not clear enough and thus, further explanation must be provided to improve the understanding of such Figure. Here, we were interested in comparing the evolution of the pressure registered at the airport of Malta, where the medicane cross over it, with the *Eulerian* evolution of the pressure obtained at the center of the cyclone. Ideally, if the center of the simulated medicane hit Malta, one would take that point fixed in Malta and obtain all the pressure values during the same period of the observations, and then compare these values with the observations. However, in our simulations most of the cyclone tracks do not cross over Malta. In those cases, and as an approximation, we took the closest grid model point associated with the cyclone track to Malta. Then, we obtain all the pressure values of that point during the same time window of the observations. In this sense, we cannot include in Fig. 15 the ensemble members that do not reproduce the cyclone, because we need a cyclone track to reproduce Fig. 15. It would not have sense to create a similar Fig. 15 showing the MSLP evolution of each ensemble member at the closest grid point to Malta because that information would not be related with each individual medicane evolution. Nevertheless, we also agree with the reviewer that one should bear in mind that these figures do not account for the full ensemble and so, special caution should be taken looking at these results.

[Figure]

*Figure 1. Example of the procedure followed to create Fig. 14. Black line corresponds to model trajectory and blue line corresponds to observed trajectory. The red point is the closest grid model point associated with the cyclone track of the model to Malta. Pressure values are obtained from this red point during the same time window of the observations.*

Following the reviewer suggestion, we have added the following sentences to clarify these points:

*"Taking into account the inherent difficulty of the models in properly predicting the intensity of TCs, the effect of assimilating conventional and RSAMV observations is explored. In this case, the lack of in-situ observations over the region where Qendresa took place is the main challenge to properly verify the cyclone's intensity forecasts in a Lagrangian sense (following the medicane evolution). Instead, we took advantage of the fact that the medicane crossed over Malta island, where METAR instruments registered a pressure drop greater than 20 hPa in 6h, reaching a minimum surface pressure value of 985 hPa. In this context, to assess the skill of the different numerical experiments, the METAR information from the Malta airport was used. **Specifically, the surface pressure evolution measured by the METAR at Malta was compared against the obtained from the ensemble members simulating the medicane for each experiment. To achieve this comparison, for each ensemble member performing the medicane, we take the evolution of the surface pressure of the closest trajectory grid point to Malta airport and then compare it with the METAR observations."***

*3. Even when using their selective validation approach, the improvements from the assimilation are not so clear. For instance on Figure 13, it is not so clear to me that overall the assimilation experiments (SYN, RSAMV, CNTR) are actually better than ensemble with data assimilation given the large bias at the end. Is it also surprising that the type of the observations assimilation (which are very diverse) do not seem to matter much.*

We understand the point of view of the reviewer, but it is important to bear in mind again the fundamental differences between NODA and the assimilation experiments (discussed in the previous point above). In addition, the large bias shown in Figure 13 at the end is likely related to the fact that in this case, after 23 hours of free simulation the impact of the observations

assimilated is reduced significantly. When assimilating observations such as reflectivity from radar, the effect last for just a couple of hours (i.e., 3-6 hours). For conventional observations such the ones assimilated in this study, the effect last longer. To clarify this point we have added the following sentence to the manuscript:

"… *After 00 UTC 8 November, the errors associated with the data assimilation experiments start to grow and the error associated with NODA start to decrease until the end of the simulation.* **Note that the large difference between NODA and the data assimilation experiments at the end of the simulation is associated with the fact that the impact of the initial conditions decreases significantly after 23-24 hours of free forecast. In other words, the model starts to "forget" the information introduced in the initial conditions from the different types of observations assimilated.** *Regarding the ensemble spread of each experiment (shaded areas in Fig. 13), it is showed that all data assimilation experiments have larger spread than the NODA. Because the number of ensemble members between NODA and the DA experiments used to compute the track error differs significantly, special caution should be taken making conclusions comparing NODA and data assimilation results."*

Regarding the concerns from the reviewer about that the type of observations assimilated do not seem to matter much, we agree, at least for this case. However, it is not surprising for us. Remember that within the subset of *in-situ* conventional observations, we have wind direction and wind speed observations. On the other hand, the RSAMV observations are just that, wind direction and wind speed, but distributed in a more homogenously way in the horizontal and in the vertical. Thus, we are assimilating basically the same type of observations but with different distributions. In addition, it is important to bear in mind that this event was shown in previous studies to be sensitive to upper-level dynamics (i.e., PV structure). So assimilating observations related to the dynamics, such as wind, are the observations that are more relevant. That's why the differences between SYN and RSAMVs are not so important, because they are assimilating basically the same wind information, but with different spatial distributions. Looking at Fig 12 and comparing SYN (Fig. 12b) with RSAMV (Fig. 12c), we can observe a significant different in the track, being RSAMVs the experiment that shows a track closer to Malta, where the cyclone crossed over.

4. *Another problem is that the manuscript does not show clearly which comparisons are against independent data and which are against assimilated data. In Figure 9, a correlation coefficient between model and maritime buoys of 0.996 is presented. If this is assimilated data, then you could easily have an correlation coefficient almost equal to 1, if you let the error variance of these observations tend to zero. But clearly, this would lead to a highly degraded forecast.*

We thank to the reviewer for pointing out this and we totally agree that we need to provide such information. The only section where we use data assimilated was to perform the different observation space diagnostics in Section 4.1 (i.e., Figs. 7-10) to check that the system was working as expected. These types of diagnostics are typically used in DA papers (e.g., Yussouf et al., 2013; Wheatley et al., 2015; Carrió et al., 2019). We have added the following sentence at the beginning of Section 4.1 to clarify this:

*"... These diagnostics and the rest of diagnosis computed in this subsection are computed using the entire set of assimilated type of observations (i.e., METARs, buoys, rawinsondes and RSAMVs)."*

Verification of the track and intensity of the medicane (Section 4.2) are performed using independent observations (i.e., observations not assimilated). To better clarify this point in the manuscript, the following sentence has been added at the beginning of Section 4.2:

"… *the potential impact of assimilating the above-mentioned observations to simulate the observed trajectory of the medicane and its intensification is investigated. Model output verification scores shown in this section will be computed using an independent set of different observations not assimilated previously, which will be (a) the surface pressure registered at the METAR located in the airport of Malta, where the cyclone crossed over it, and (b) the approximated medicane track obtained by visual inspection of infrared satellite imagery.*"

**Specific comments:**

- *Line 55: "Although both methods are slightly different and contain different types of errors associated, the overall information drawn from them has been found to be equivalent (Migliorini, 2012). From these reasons, in this study only satellite-derived products will be considered." The cited study Migliorini (2012) does not show that the assimilation for both methods is an all cases equivalent. It rather defines two testable conditions, under which both approaches lead to equivalent results: "(i) the radiance observation operator needs to be approximately linear in a region of the state space centered at the retrieval and with a radius of the order of the retrieval error; and (ii) any prior information used to constrain the retrieval should not underrepresent the variability of the state, so as to retain the information content of the measurements." The author should check that two conditions deduced by Migliorini (2012) are verified before stating that both methods are equivalent.*

This sentence has been modified to the following:

"" → "*To avoid dealing with nonlinearities associated with the assimilation of radiance observations using RTM, only satellite-derived products will be considered in this study.*"

- *Line 92: "Among the different available medicanes, the so called Qendresa, which took place southern Sicily between 7-8 November 2014 (Carrió et al., 2017) and was poorly forecasted, was selected to perform this study. More precisely, the correct prediction of both the northward loop trajectory followed by Qendresa and its intensification still remain a major challenge for most current numerical weather models." Can you be more specific which models did a poor forecast and show a figure of the forecast and actual and predicted path (and intensity)?*

Qendresa is a well-know medicane in the weather forecast community in the Mediterranean that has been characterized to be very difficult to predict in terms of intensity and trajectory. As the present study is a continuation of our last study on this medicane (Carrió et al., 2017), we strongly believe that adding a new figure and more details about the models used in previous studies in the present manuscript could distract the reader attention of the main line of this study and will not provide useful information. In addition, it is not desirable to add more figures in this manuscript because the large number of them in the present version. However, we agree with the reviewer that we need to add more information about this and

adding references could be the best way of providing such information. We have modified the original sentence as follows:

"~~Among the different available medicanes, the so-called Qendresa, which took place southern Sicily between 7-8 November 2014 (Carrió et al., 2017) and was poorly forecasted, was selected to perform this study. More precisely, the correct prediction of both the northward loop trajectory followed by Qendresa and its intensification still remain a major challenge for most current numerical weather models.~~" → "Among the different available medicanes, the so called Qendresa, which took place southern Sicily between 7-8 November 2014 and was poorly forecasted, was selected to perform this study. More precisely, the correct prediction of both the northward loop trajectory followed by Qendresa and its intensification remain a major challenge for most current numerical weather models (e.g., Carrió et al., 2017; Cioni et al., 2018; Pytharoulis, 2018; Noyelle et al., 2019; Bouin et al., 2020)."

**References:**

Carrió, D. S., Homar, V., Jansa, A., Romero, R., & Picornell, M. A. (2017). Tropicalization process of the 7 November 2014 Mediterranean cyclone: Numerical sensitivity study. Atmospheric Research, 197, 300-312.

Pytharoulis, I. (2018). Analysis of a Mediterranean tropical-like cyclone and its sensitivity to the sea surface temperatures. Atmospheric Research, 208, 167-179.

Cioni, G., Cerrai, D., & Klocke, D. (2018). Investigating the predictability of a Mediterranean tropical-like cyclone using a storm-resolving model. Quarterly Journal of the Royal Meteorological Society, 144(714), 1598-1610.

Noyelle, R., Ulbrich, U., Becker, N., & Meredith, E. P. (2019). Assessing the impact of sea surface temperatures on a simulated medicane using ensemble simulations. Natural Hazards and Earth System Sciences, 19(4), 941-955.

Bouin, M. N., & Lebeaupin Brossier, C. (2020). Impact of a medicane on the oceanic surface layer from a coupled, kilometre-scale simulation. Ocean Science, 16(5), 1125-1142.

- *Line 157: "These simulations used a multi-scale ensemble system based on two one-way nested domains to better account for meso- and storm-scale processes involved in the genesis and evolution of Qendresa (Fig. 2)." Why only one-way nesting is used here (WRF supports two-way nesting if I am not mistaken)?*

The reviewer is right, the WRF support two-way nesting. However, from the wide experience of my former research Group, we noticed that for this kind of configuration where the inner domain is centered over a quite large parent domain, such the one used in this study, no differences are appreciated apart from the fact that the two-way nesting takes more time to run the simulations. In cases where the boundary conditions are close to the inner domain could be more beneficial to use the two-way nesting.

- *Line 230: "Following similar studies (e.g., Romine et al. (2013); Yussouf et al. (2015); Carrió and Homar (2016)), the observational error values used here for the conventional observations are: 0.75 K for the temperature, 0.75 K for the dew point temperature, 0.75 m s −1 for the wind speed and 0.75 hPa for the pressure." In Romine et al. (2013), they use "NCEP statistics" for temperature, Lin and Hubbard (2004) for the dew point temperature, 1.75 m/s for E–W, N–S winds (Buoy and ship reports) and 1 hPa for altimeter (also Buoy and ship reports). Also AMV errors are 50% NCEP statistics in Romine et al. (2013). This seems to me (not a meteorology expert) quite different from the fixed value approach here. In Yussouf et al. (2015), the used observation errors are described as: The assumed observation errors are the same as in Table 3 of Romine et al. (2013) except for METAR and marine temperature (1.75 K), METAR altimeter (0.75 hPa), and marine altimeter (1.20 hPa). I don't understand how the citations are used to support the choice of these parameter values (except for the METAR altimeter) which are crucial for data assimilation (as also noted by the author).*

We thank the reviewer for pointing out this misunderstanding. We initially based our choices to studies dealing with similar problems, such as Romine et al. (2013) and Yussouf et al. (2015). However, we end using the same values used in Carrió et al. (2018), where observational error used were analogous to Table 3 in Romine et al. (2013) with minor exceptions: METAR altimeter (1.5 hPa), marine altimeter (1.20 hPa) and METAR and marine temperature (1.75 K).

Following the reviewer comment, we have modified the original sentence as follows:

"→ *"The observational error values used in this study for the conventional observations are analogous to Table 3 in Romine et al., (2013) with minor exceptions: METAR altimeter (1.5 hPa), marine altimeter (1.20 hPa) and METAR and marine temperature (1.75 K)."*

- *Figure 7 and 9: there are several assimilation runs introduced in the previous section. It is not clear to me which assimilation experiment is presented in Figure 7. Also, I think the author should take more clearly which comparisons are performed against independent data and which use dependent data (used in the analysis). Since for figure 9, a correlation coefficient of 0.996 is achieved for the posterior estimate, I suspect that this is comparison with dependent observation. Much more interesting would be validation with independent data: for example if you assimilated only RSAMV data, do you also improve compared to METARs, rawinsondes and buoys?*

We thank the reviewer for this comment, and we agree that no information about which experiment comes from these results is present. Fig's 7-10 are obtained from the CNTRL experiment. As we stated previously, such figures were performed using dependent observations, which is the usual way of verifying that the DA system is working as we expect. Then, to verify the forecasts results, we use independent observations. This methodology is very common in DA studies (see for example Romine et al. 2013; Wheatley et al, 2015; Jones et al, 2016; Yussouf et al., 2015).

In order to provide this information to the reader we have added the following sentence at the beginning of the Section 4.1:

*"To quantitatively assess the data assimilation performance during the 12-h data assimilation window, the following widely-used observation-space diagnostics (Yussouf et al., 2013; Wheatley et al., 2015; Carrió et al., 2019) are computed before and after each hourly data assimilation cycle using the background and EnKF analysis model states, **from the CNTRL experiment**, mapped to the observation locations"*

**References:**

*Wheatley, D. M., K. H. Knopfmeier, T. A. Jones, G. J. Creager, 2015: Storm-Scale Data Assimilation and Ensemble Forecasting with the NSSL Experimental Warn-on-Forecast System. Part I: Radar Data Experiments. Weather and Forecasting, 30, 1795–1817, doi:*10.1175/WAF-D-15-0043.1

*Jones, T. A., K. Knopfmeier, D. Wheatley, G. Creager, P. Minnis, R. Palikonda, 2016: Storm-Scale Data Assimilation and Ensemble Forecasting with the NSSL Experimental Warn-on-Forecast System. Part II: Combined Radar and Satellite Data Experiments. Weather and Forecasting, 31, 297–327, doi:*10.1175/WAF-D-15-0107.1

*Yussouf, N., D. C. Dowell, L. J. Wicker, K. H. Knopfmeier, D. M. Wheatley, 2015: Storm-scale Data Assimilation and Ensemble Forecasts for the 27 April 2011 Severe Weather Outbreak in Alabama. Monthly Weather Review, 143, 3044–3066.*

- *Figures: text on Figure 7 and 8 is too small .*

The figures and labels have been enlarged.

- *Line 370: "In fact, for the SYN experiment only 17/36 ensemble members generate a small-scale isolated cyclone, while in the RSAMV, a reduced number of members simulate cyclones (16/36), and finally in the CNTRL experiment, this number is increased to 21/36." How many ensemble members generate a cyclone for the simulation NODA and how many ensemble members in NODA are used for validation later on?*

As we stated on the first point of the reviewer above, 29/36 ensemble members generate a cyclone for NODA. For the same reasons I explained above, 29 members were used in the verification later. This information is now present in the last version of the manuscript.

- *Line 375: "Taking into account this, we have also represented the best track simulated by the different experiments in comparison with the trajectory observed by satellite imagery." How do you define "best" here?*

We appreciate the referee comment about this point. Here, to simplify the problem, we have defined "best" in a subjective way, just comparing visually the different tracks for each ensemble member with the observed track. We were interested to obtain tracks with two main properties: (a) tracks that crossed over Malta and (b) tracks that show the loop-pattern at the end of the simulation. In order to avoid misunderstanding, we have rephrased the sentence in the following way:

*"Taking into account this, we have also represented, for each experiment, the track from the ensemble member that subjectively resembles the most the medicane trajectory observed by satellite imagery. For this purpose, we seek for two main features: (a) the medicane center should cross as close as possible to Malta island and ideally (b) the track of the medicane should show signals of a loop-ending on the eastern coast of Sicily."*

- *Line 386: "and the error associated with NODA start to decrease until the end of the simulation." and Figure 13: Overall the assimilation did not improve a lot compared to the simulation without data assimilation. During 17:00 - 23:00 November 7, the assimilation run has an error which is about 50 km lower, however during the end the error is about 50-100 km larger.*

The fact that the error of the assimilation experiments is bigger than NODA has already been explained above and it was related to the fact that after 23 hours of free model run, the information introduced to the estimate of the initial conditions through the assimilation is not affecting the forecast anymore. The time window in which the assimilation has effects on the forecasts depend on the type of observations. For instance, the impact of assimilating, reflectivity observations, lasts between 3-6 hours. Beyond that time, the forecast "forgets" about the data that was assimilated.

- *Figure 13: "ensemble spread (shaded areas) of track error (km)": not all ensemble members present a cyclone. How is this taken into account? If in an ensemble, only one member would have represented a cyclone, would this correspond to a spread of 0 ?*

The members that do not present a cyclone are not considered. In the case of having just one member representing a cyclone, Fig. 13 would show a single line. However, we totally agree that Fig. 13 alone could be misinterpreted. For this reason, we have added a line in the manuscript that reminds the reader that those results are obtained considering different number of ensemble members. This point has already been solved previously.

*Figure 14: this analysis considers only the center of the medicane; if an ensemble would have a small but consistent biais the medine track, it would result in a very low score when computing the PCC. It would be more useful to compute a probability map for the wind exceeding a given threshold.*

This figure was deleted after revision of reviewer #1.

- *Intensity (Figure 15): all experiments underestimate the intensity (indicated by the minimum pressure). But the NODA experiment most realistic mean (in terms of minimum pressure) and a significant fraction of the ensemble members for NODA show the right intensity (but with a time shift) while only few ensemble members with data assimilation for the have the correct intensity (but with a better timing). Timing better and better minimum pressure?*

Although NODA seems to have more ensemble members showing the right intensification (with a time shift) it also shows that a lot of ensemble members depicts shallow cyclones that are totally misplaced from the observed maximum of intensification. For instance, Fig. 15a shows that after 8 November 00 UTC a significant number of ensemble members depicts shallow cyclones. This is totally wrong. However, when we assimilate the different observations, we avoid such spurious cyclones. Again, is difficult to account for all ensemble

members, including the ones not showing a cyclone because the verification scores we are using requires having a cyclone. This points has already been solved in some of the above comments.To account for this problem, we have added a sentence in the manuscript that reminds the reader that the number of ensemble members used for each experiment is different, and so, general conclusions cannot be obtained.

**Minor issues:**

*Line 55: "From these reasons" -> "For these reasons"*

Done

*Line 104: Add citation for EnKF when mentioning it the first time (after the abstract). There are many variants of the EnKF. The reference is only included in section 3.3.*

Done

*Line 205: "In other words, one spectral channel can identify the same wind observation that another channel can identify. However, is not common that both different channels provides precisely the same value of such observation." The formulation is a bit awkward (besides the spelling issues "However, is" -> "However, it is", "provides" -> "provide"). Please rephrase.*

Done. "For instance, spectral channel number 1 could identify the same wind observation that channel number 4 could identify. However, it is not common that both channels provide precisely the same value of such observation, they are slightly different."

*Line 317: rmsi (and legend of plot 7) -> RMSE*

Done

---

## Author Response (AR2)

**The University of Melbourne**
Diego Saúl Carrió Carrió
School of Geography, Earth and Atmospheric Sciences and
Centre of Excellence for Climate Extremes
Parkville, 3010
Victoria, AUSTRALIA
diego.carriocarrio@unimelb.edu.au
Melbourne, January 12, 2023

Dr. Joanna Staneva and Piero Lionello
Editor and Executive Editor– Natural Hazards and Earth System Sciences

Dear Joanna Staneva,

Please find attached the revised version of the manuscript **NHESS-2022-58** entitled ***"Challenges assessing the effect of AMVs to improve the predictability of a medicane weather event using the EnKF. Storm-scale analysis and short-range forecast"***.

I really appreciate the comment of the reviewer to improve the quality of the manuscript. I have followed the comments that were suggesting making bigger the axes of many of the plots. I have enlarged all the figures of the manuscript and now they can be read without problems. I thank both of the reviewers for their comments and I believe the manuscript is now greatly improved both in terms of clarity and readability.

Dear Piero Lionello,

I really appreciate your comments to improve the manuscript. Please find attached my answers to your comments.

With best regards,

Diego Carrió Carrió

**ANSWERS TO THE EXECUTIVE EDITOR**

*Dear Dr. Carrió:*

*Before accepting the manuscript (subject to technical correction) I have some suggestions to improve mainly title, abstract and other details.*

*- Please, make your article title more attractive by avoiding abbreviations. A tentative suggestion of mine is "The improvement of the predictability of the Qendresa medicane by the assimilation of in-situ observations and atmospheric motion vectors". Feel fully free to reject or modify this suggestions for the title, but I insist that you at least avoid abbreviations.*

I really appreciate your suggestion and I agree with you that I could make the title more appealing. I have changed the title of the manuscript to the following:

"*Improving the predictability of the Qendresa medicane by the assimilation of Conventional and Atmospheric Motion Vector observations. Storm-scale analysis and short-range forecast.*"

*- Line 2 "invaluable amount" means "an amount valuable beyond estimation. I think this is not correct, as estimates are possible, though they are necessarily not exact. Please delete or choose a more appropriate qualification*

Thanks for noticing that. I have modified the sentence as follows:

"*Coastal population in the Western Mediterranean basin is frequently affected by high-impact weather events that produce huge economic and human losses.*"

*- Line 4 "to our community": In this context it is not clear which community is. Please rewrite and specify*

I agree that this part of the sentence could be misleading. I have modified the sentence as follows:

"*The accurate prediction of this kind of events still remains a key challenge to the weather forecast community, mainly because of (i) the errors in the initial conditions, (ii) the lack of accuracy of modelling micro-scale physic processes and (iii) the chaotic behavior inherent to numerical weather prediction models.*"

*- Line 5 I think "of" is needed between modelling and microscale*

I modify the sentence as follows:

"*... the lack of accuracy **of** modelling micro-scale physic processes and ...*"

*- Line 4 -5 I think the use of definite articles in this sentence is not according to English style*

I have changed the sentence as follows:

*"... mainly because of (i) errors in the initial conditions, (ii) lack of accuracy of modelling micro-scale physic processes and (iii) chaotic behavior inherent to numerical weather prediction models..."*

*- Line 6 "in particular" seems redundant to me*

I have deleted *"in particular"* from the sentence and now it can be read as:

*"The 7th November 2014 Qendresa medicane, that took place over the Sicilian channel affecting the Islands of Lampedusa, Pantelleria and Malta was selected for this study because of its extremely low predictability behavior in terms of its track and intensity."*

*- Line 11 "comma" after "thus" is redundant*

Agree. I have deleted the "*comma*" after "*thus*".

*- Maps in figures 15, 16 and 17 are very difficult to be read. Is it possible to reduce the number of contours?... beside increasing the size of labels as already commented by the handling editor*

I believe that in the last version of the paper, these figures can be read without problems.

*- Line 523 "a rare type" is ambiguous (are medicanes rare or this specific medicane has some rare characteristic?... I suggest to delete "a rare type"*

This is an adjective that other authors have already used in their papers and it is accepted in the community. However, I will follow your suggestion and I will delete it from the sentence. Instead, the sentence is now written as follows:

"*A particular type of Mediterranean cyclone has drawn the attention of the meteorological scientific community.*"